# Minimisation of Quasar-Convex Functions Using Random Zeroth-Order Oracles

**Amir Ali Farzin**                                                    *amirali.farzin@anu.edu.au*
*School of Engineering*
*Australian National University*

**Yuen-Man Pun**                                                      *yuenman.pun@unimelb.edu.au*
*Department of Electrical and Electronic Engineering*
*University of Melbourne*

**Philipp Braun**                                                       *philipp.braun@anu.edu.au*
*School of Engineering*
*Australian National University*

**Iman Shames**                                                        *iman.shames@unimelb.edu.au*
*Department of Electrical and Electronic Engineering*
*University of Melbourne*

**Reviewed on OpenReview:** *https://openreview.net/forum?id=rRp9zZBKkZ*

## Abstract

This paper explores the performance of a random Gaussian smoothing zeroth-order (ZO) scheme for minimising quasar-convex (QC) and strongly quasar-convex (SQC) functions in both unconstrained and constrained settings. For the unconstrained problem, we establish the ZO algorithm's convergence to a global minimum along with its complexity when applied to both QC and SQC functions. For the constrained problem, we introduce the new notion of proximal-quasar-convexity and prove analogous results to the unconstrained case. Specifically, we derive complexity bounds and prove convergence of the algorithm to a neighbourhood of a global minimum whose size can be controlled under a variance reduction scheme. Beyond the theoretical guarantees, we demonstrate the practical implications of our results on several machine learning problems where quasar-convexity naturally arises, including linear dynamical system identification and generalised linear models.

## 1 Introduction

In this paper, we study a minimisation problem of the form

$$\min_{x \in \mathcal{X}} \quad f(x) \tag{1}$$

where $f : \mathbb{R}^n \to \mathbb{R}$ is continuously differentiable, integrable, possibly non-convex and bounded below, and $\mathcal{X} \subset \mathbb{R}^n$ is a non-empty convex set. Such problems arise routinely in machine learning and control.

To solve problem (1), several classes of optimisation methods have been proposed in the literature. Most existing methods rely on access to the function's gradient, limiting their applicability in many real-world scenarios. For instance, in many practical systems in machine learning, one can only observe the input and output of a deep neural network (DNN) without access to its internal configurations, such as the network structure and weights. Optimisation methods that rely solely on the (noisy) evaluations of the objective function are commonly referred to as zeroth-order (ZO) or derivative-free frameworks (Rios & Sahinidis, 2013; Audet & Hare, 2017). These schemes have attracted increasing interests in recent years due to their

success in solving black-box problems arising in machine learning and signal processing, such as automated backpropagation in deep learning (DL) (Liu et al., 2020b) and evaluating the adversarial robustness of DL networks (Goodfellow et al., 2014), where explicit expressions of gradients are expensive to compute (Cartis et al., 2010) or even unattainable (Maass et al., 2021). ZO methods provide a solution to these challenges by estimating gradients using function evaluations.

Multiple ZO algorithms have been developed to solve various optimisation problems, for example, methods inspired by evolutionary strategies (Moriarty et al., 1999), methods based on direct search (Vicente, 2013) (Anagnostidis et al., 2021), and those that rely on unbiased variance-bounded approximators of the gradient (Ghadimi et al., 2016). Recently, Nesterov & Spokoiny (2017) proposed and analysed a random ZO oracle based on *Gaussian smoothing*. This approach yields a gradient estimate of a point $x$ by computing the function values of the point $f(x)$ and the function value of a point in a neighbourhood of $x$ based on Gaussian sampling. As an unbiased estimator of the gradient of the Gaussian smoothed function, this zeroth-order method is gaining attention in optimisation because of the desirable properties of the Gaussian smoothed function. For example, the Gaussian smoothed function is shown to inherit the convexity and Lipschitzness of the original function and it possesses a Lipschitz gradient as long as the original function is globally Lipschitz (Nesterov & Spokoiny, 2017). Existing results indeed demonstrate its efficacy when applying different optimisation problems. However, existing convergence guarantees for Gaussian-smoothing ZO methods are largely restricted to convex, strongly convex, or generic nonconvex settings (e.g., stationarity), and do not exploit structural properties such as quasar-convexity that have recently been identified in several learning problems. Nesterov & Spokoiny (2017) shows that their proposed algorithm, in expectation, converges to an $\epsilon$-optimal point (i.e., $f(x) - f(x^*) \leq \epsilon$) in $\mathcal{O}(n\epsilon^{-1})$ iterations and $\mathcal{O}(n\log(\epsilon^{-1}))$ iterations when applied to a smooth convex function and a smooth strongly convex function, respectively. When applied to a non-smooth convex function, the algorithm converges to an $\epsilon$-optimal point in $\mathcal{O}(n^2\epsilon^{-2})$ iterations. The authors analyse smooth but not necessarily convex cost functions and show convergence of their algorithm to an $\epsilon$-stationary point (i.e., $\|\nabla f(x)\| \leq \epsilon$) in $\mathcal{O}(n^2\epsilon^{-2})$ iterations. Considering non-convex functions, Farzin & Shames (2024) focus on solving the minimisation problem with smooth objective functions satisfying Polyak-Łojasiewicz inequality. It is shown that their proposed methods converge to an $\epsilon$-optimal point in $\mathcal{O}(n\epsilon^{-1})$ iterations in the unconstrained case and they converge to a neighbourhood of the set of $\epsilon$-optimal points in $\mathcal{O}(\epsilon^{-1})$ iterations in the constrained case, where the size of the neighbourhood is proportional to the variance of the norm of the error of the gradient estimate constructed by the ZO method. Pougkakiotis & Kalogerias (2023) show that the proximal Gaussian-smoothing ZO oracle converges to an $\epsilon$-stationary point of the smoothed objective function in $\mathcal{O}(\sqrt{n}\epsilon^{-4})$ iterations, when applied to a non-smooth weakly-convex function. In Farzin et al. (2025b), the authors consider solving the offline and online minimisation of submodular cost functions leveraging a Gaussian ZO framework and show that their method finds an $\epsilon$-optimal point in $\mathcal{O}(n^2\epsilon^{-2})$ iterations. In Vicente (2013), the author analyses unconstrained non-convex minimisation and, by leveraging the direct search method, the author shows the convergence to an $\epsilon$-stationary point in $\mathcal{O}(n^2\epsilon^{-2})$ iterations.

In this paper, we study the potential of the Gaussian-smoothing ZO oracle applied to a class of non-convex functions that satisfy *quasar-convexity*. Quasar-convexity is a weaker notion than star-convexity and has been found in a number of problems that are closely related to DL. For example, Hardt et al. (2018) showed that the problem of learning linear dynamical systems, which is known to be closely related to recurrent neural networks, satisfies quasar-convexity under some mild assumptions. Wang & Wibisono (2023) shows that generalised linear models with activation functions including leaky ReLU, quadratic, logistic, and ReLU functions, satisfy quasar-convexity. Results from a number of recent research works also suggest that neural networks may satisfy some kind of quasar-convexity (Lin et al., 2024; Zhou et al., 2018). Moreover, we are interested in exploring the convergence of the ZO oracle when applied to strong quasar-convex functions. While it is known that the Polyak-Łojasiewicz (PŁ) (Polyak, 1964) condition is weaker than strong quasar-convexity (Hinder et al., 2020), it can be shown that a function which is PŁ (or satisfying quadratic growth condition) and quasar-convex is strongly quasar-convex (Wang & Wibisono, 2023, Lemma 7). Therefore, a number of learning problems, including linear residual networks (in large regions of parameter the space) (Hardt & Ma, 2017) and entropy regularised policy gradient optimisation in a class of reinforcement learning problems (Mei et al., 2020), might be of interest. In this sense, quasar-convexity captures a nontrivial subset of nonconvex objectives that are both practically relevant in modern machine learning (ML) and amenable to sharper guarantees than those available for general nonconvex functions.

Crucially, some of the machine learning problems where quasar-convexity arises are precisely those where gradient information is expensive, unreliable, or unavailable—making ZO methods natural algorithms of choice. For instance, in a linear dynamical system identification, the objective is quasar-convex under mild assumptions (Hardt et al., 2018), yet computing exact gradients requires backpropagation through time (BPTT), which suffers from large and vanishing gradients in long sequences (Pascanu et al., 2013). A ZO approach sidesteps BPTT entirely by relying only on trajectory rollouts, which are cheap to obtain in simulation or from physical systems (Maass et al., 2021). Similarly, when training or fine-tuning large-scale models where only input-output data is available, such as querying a black-box API of a deep neural network, ZO methods are the only viable option (Liu et al., 2020b; Chen et al., 2024), and the underlying loss landscapes of such models have been observed to exhibit star-convex or quasar-convex structure along optimisation paths (Zhou et al., 2018; Lin et al., 2024). In reinforcement learning, policy optimisation under entropy regularisation satisfies conditions closely related to strong quasar-convexity (Mei et al., 2020; Cen et al., 2022), while the policy gradient itself must often be estimated from sampled rollouts without access to an analytic gradient, effectively constituting a ZO setting (Moriarty et al., 1999). These examples illustrate that ZO quasar-convex optimisation is not merely a theoretical intersection of two separate research threads, but a problem class that naturally emerges in modern machine learning practice.

These applications have spurred exciting algorithmic design and analysis for functions that satisfy quasar-convexity. For example, Hinder et al. (2020) develop an accelerated first-order algorithm and show its iteration complexity for both strong quasar-convex and quasar-convex functions. In Guminov et al. (2017) the convergence of Nemirovski's conjugate gradients when applied to functions satisfying quasar-convexity and quadratic growth condition is studied. Nesterov et al. (2018) proposes an accelerated gradient method and shows the convergence when minimising quasar-convex functions. Additionally, Jin (2020) studies the convergence of stochastic gradient descent when applied to both strong quasar-convex and quasar-convex functions. However, to the best of our knowledge, all the mentioned papers on (strong) quasar-convex function minimisation are leveraging first-order algorithms, while our work is the first that studies the convergence of a random zeroth-order method for (strong) quasar-convex functions.

**Contributions** In this paper, we study the random ZO oracle in Nesterov & Spokoiny (2017) to solve the minimisation problem (1) for a class of quasar-convex objective functions. Our main contributions are as follows.

*(i) Zeroth-order convergence under quasar-convexity.* We first consider the unconstrained setting of problem (1). We show that Algorithm 1 converges to an $\epsilon$-optimal point in $\mathcal{O}(n\epsilon^{-1})$ iterations when applied to quasar-convex functions and in $\mathcal{O}(n\log(\epsilon^{-1}))$ iterations when applied to strongly quasar-convex functions. Despite the non-convexity, these rates match the order of the best known guarantees for convex and strongly convex functions.

*(ii) Proximal quasar-convexity and constrained problems.* For the constrained setting of problem (1), we introduce a new notion, called *proximal quasar-convexity*, as an analogue to quasar-convexity in non-smooth optimisation. We show that, when the function satisfies proximal quasar-convexity (resp. strong proximal quasar-convexity), Algorithm 1 converges to a neighbourhood of the set of $\epsilon$-optimal points in $\mathcal{O}(n\epsilon^{-1})$ iterations (resp. in $\mathcal{O}(n\log(\epsilon^{-1}))$ iterations). We further show that the size of this neighbourhood can be reduced to values arbitrarily close to zero using a variance reduction scheme.

*(iii) Empirical evidence on ML-motivated problems.* Although ZO methods are often regarded as inferior to first-order methods in terms of sample efficiency, our numerical results show that Algorithm 1 can perform comparably to, and in some regimes better than, gradient descent when learning a linear dynamical system, and behaves competitively on other quasar-convex ML tasks such as generalised linear models and support vector machines. These experiments illustrate that the structural assumptions studied in our theory are not only analytically convenient but also arise in practical ML problems. A possible explanation of the better performance of Algorithm 1 in comparison with GD, in test cases like learning linear dynamical systems, is given in Appendix F.

**Outline:** In Section 2, we outline the problems of interest and introduce the Gaussian-smoothing ZO oracle. Section 3 presents the main results on convergence and iteration complexities of the Gaussian-smoothing

ZO oracle when applied to (strongly) quasar-convex functions in the unconstrained setting of problem (1) and to (strongly) proximal quasar-convex functions in the constrained setting when $\mathcal{X} \neq \mathbb{R}^n$ in (1). Section 4 offers illustrative examples, numerically confirming the performance and convergence properties of the proposed algorithm. Lastly, we conclude our paper and discuss potential future research directions in Section 5. Auxiliary lemmas, proofs of the main theorems, complementary material and additional numerical experiments can be found in the appendix.

**Notation:** In this paper, $\mathbb{R}^n$, $n \in \mathbb{N}$, denotes the $n$-dimensional Euclidean space with $\langle \cdot, \cdot \rangle$ as the inner product. We denote the set of positive real numbers with $\mathbb{R}_{>0}$. We use $\|\cdot\|$ to denote the Euclidean norm of its argument if it is a vector and the corresponding induced operator norm if the argument is a matrix. We denote the weighted norm of a vector by $\|\cdot\|_M$, where $\|u\|_M = \langle Mu, u \rangle^{1/2}$, $u \in \mathbb{R}^n$, and $M$ is a positive definite matrix. The projection operator to a closed convex set $\mathcal{X} \subset \mathbb{R}^d$, is defined as $\mathrm{Proj}_{\mathcal{X}}(z) \overset{\text{def}}{=} \arg\min_{x \in \mathcal{X}} \|x - z\|^2$. The expectation operator with respect to a random variable $u$ is denoted by $E_u[\cdot]$. For $k \in \mathbb{N}$, we denote by $\mathcal{U}_k = \{u_1, \ldots, u_k\}$ a set comprising of independent and identically distributed random vectors $\{u_k\}_{k>0}$. The conditional expectation over $\mathcal{U}_k$ is denoted by $E_{\mathcal{U}_k}[\cdot]$. The diameter of a bounded set $\mathcal{X}$ is denoted by $d_{\mathcal{X}}$ and is equal to $d_{\mathcal{X}} = \sup\{\|x_1 - x_2\| : x_1, x_2 \in \mathcal{X}\}$.

## 2 Problems of Interest And Framework

In this paper, we study the performance of a random Gaussian smoothing ZO method applied to minimisation problems involving (strong) quasar-convex functions in unconstrained and constrained settings. Our goal is to understand if such a generic black-box optimisation scheme can enjoy the same type of guarantees as gradient-based methods by exploiting the quasar-convex structure that appears in several modern ML models. The Gaussian smoothed version of a continuous function $f : \mathbb{R}^n \to \mathbb{R}$, denoted by $f_\mu : \mathbb{R}^n \to \mathbb{R}$, is defined as:

$$f_\mu(x) = \frac{1}{\kappa} \int_{\mathbb{R}^n} f(x + \mu u) e^{-\frac{1}{2}\|u\|_B^2} \mathrm{d}u, \qquad \kappa = \int_{\mathbb{R}^n} e^{-\frac{1}{2}\|u\|_B^2} \mathrm{d}u = \frac{(2\pi)^{n/2}}{[\det B]^{\frac{1}{2}}} \tag{2}$$

where $u \in \mathbb{R}^n$ is sampled from a mean Gaussian distribution with a positive definite correlation operator $B^{-1} \in \mathbb{R}^{n \times n}$ and $\mu > 0$ is the smoothing parameter. The random oracle $g_\mu : \mathbb{R}^n \to \mathbb{R}^n$ is defined as (Nesterov & Spokoiny, 2017, Section 3)

$$g_\mu(x) = \frac{f(x + \mu u) - f(x)}{\mu} Bu, \tag{3}$$

where $u \in \mathbb{R}^n$ and $B \in \mathbb{R}^{n \times n}$ are defined above. In Nesterov & Spokoiny (2017), it is shown that $g_\mu$ is an unbiased estimator of $\nabla f_\mu$; i.e., $\nabla f_\mu(x) = E_u[g_\mu(x)]$. We use this random ZO oracle to update our estimate of the solution to (1). To ensure the feasibility of the generated points in the constrained case, we leverage a projection step.

**Remark 1.** *In the following, we use the function $g_\mu(\cdot)$ to approximate the derivative of a continuously differentiable function $f$. While in this case the smoothing step (2) may not be necessary, the more general gradient estimator (3) is valid in applications where function evaluations of $f(\cdot)$ are cheap compared to an explicit calculation and evaluation of the gradient $\nabla f(\cdot)$, or explicit calculation and evaluation of the gradient $\nabla f(\cdot)$ may not possible, as motivated in the introduction.*

**Remark 2.** *In the rest of the paper, we assume that $B \in \mathbb{R}^{n \times n}$ is selected as the identity matrix $B = \mathbb{I}$, as is generally assumed in the literature on Gaussian smoothing (Balasubramanian & Ghadimi, 2022; Farzin & Shames, 2024; Chen et al., 2024). However, we note that, extensions for the general choice of $B$ are possible, following the discussions in Farzin et al. (2025a).*

The algorithm to solve problem (1) in this paper is summarised in Algorithm 1, where $x_0$ is the initial guess, $\mu > 0$ is the smoothing parameter, $t \in \mathbb{N}$ is the number of samples in each iteration, $h_k > 0$ is the step size, and $N \in \mathbb{N}$ is the number of iterations.

Algorithm 1 is the natural projected variant of the Gaussian-smoothing ZO method of Nesterov & Spokoiny (2017): at each iteration, it averages $t \in \mathbb{N}$ independent two-point function evaluations to construct an

---

**Algorithm 1** Random Min (RM)

---

1: **Input:** $x_0 \in \mathcal{X}$, $N \in \mathbb{N}$, $\{h_k\}_{k=1}^N$, $\mu > 0$, $t \in \mathbb{N}$
2: **for** $k = 1$ to $N$ **do**
3:     Sample $u_k^1, \ldots, u_k^t$ from $\mathcal{N}(0, \mathbb{I})$
4:     Calculate $g_\mu^1(x_k), \ldots, g_\mu^t(x_k)$ using $u_k^0, \ldots, u_k^t$ and (3).
5:     $g_\mu(x_k) = \frac{1}{t} \sum_{i=1}^t g_\mu^i(x_k)$
6:     $x_{k+1} = \mathrm{Proj}_\mathcal{X}(x_k - h_k g_\mu(x_k))$
7: **end for**
8: **return** $x_N$

---

approximate gradient direction and then performs a projected update. In our analysis, the batch size $t$, smoothing parameter $\mu > 0$ and step sizes $\{h_k\}$ will be chosen to balance the bias introduced by smoothing and the variance of the random oracle, leading to dimension- and accuracy-dependent iteration bounds under (proximal) quasar-convexity. We aim to investigate the performance of Algorithm 1 when applied to quasar-convex functions. First, we consider the unconstrained setting of problem (1); i.e., $\mathcal{X} = \mathbb{R}^n$. Then, we analyse the performance in the constrained setting of problem (1); i.e., $\mathcal{X} \subset \mathbb{R}^n$, $\mathcal{X} \neq \mathbb{R}^n$. Before proceeding further, we need to define quasar-convexity.

**Definition 1** (Quasar-convex function). *Let $f : \mathbb{R}^n \to \mathbb{R}$ be a continuously differentiable function, $\gamma \in (0, 1]$, $\beta > 0$, and $x^*$ be a minimiser of $f$. We say that $f$ is $\gamma$-quasar-convex with respect to $x^*$ if for all $x \in \mathbb{R}^n$,*

$$f(x^*) \geq f(x) + \frac{1}{\gamma} \langle \nabla f(x), x^* - x \rangle. \tag{4}$$

*The function $f$ is $\beta$-strongly-$\gamma$-quasar-convex with respect to $x^*$ if for all $x \in \mathbb{R}^n$,*

$$f(x^*) \geq f(x) + \frac{1}{\gamma} \langle \nabla f(x), x^* - x \rangle + \frac{\beta}{2} \|x^* - x\|^2. \tag{5}$$

Quasar-convexity is found in a number of optimisation problems that are closely related to DL or ML (Hardt et al., 2018; Wang & Wibisono, 2023; Hardt & Ma, 2017; Mei et al., 2020). It also possesses benign properties for the ease of analysis. For example, it can be seen that any stationary point of a quasar-convex function is its global minimum and the minimiser of a strongly-quasar-convex function is unique. Compared to general nonconvex objectives, quasar-convex functions therefore form an intermediate class. They preserve global optimality properties reminiscent of convexity, while still capturing nontrivial nonconvex ML models as discussed in Section 1.

Next, we consider the constrained setting of problem (1). Suppose that $\mathcal{X} \subset \mathbb{R}^n$ is a non-empty compact convex set with diameter $d_\mathcal{X}$. Since quasar-convexity is well-defined only for unconstrained problems with the objective being differentiable (Hinder et al., 2020), we reformulate the constrained problem to a non-differentiable unconstrained problem and introduce the notion of *proximal quasar-convexity*. This formulation naturally covers composite objectives of the form "loss + convex regulariser/indicator", which are ubiquitous in ML. To introduce the concept of proximal quasar-convex functions we first need to define the function

$$Q_l(x, a) = \frac{1}{a}(x - \mathrm{Prox}_l(x - a\nabla f(x))), \tag{6}$$

where $a > 0$ denotes a positive scalar and

$$\mathrm{Prox}_l(x) = \arg\min_z (\|z - x\|^2 + l(z) - l(x)) \tag{7}$$

defines the proximal operator corresponding to a convex extended real-valued function $l : \mathbb{R}^n \to \mathbb{R} \cup \{\infty\}$.

**Definition 2** (Proximal $\gamma$-quasar-convex Functions). *Let $\gamma \in (0, 1]$ and $\beta > 0$. Consider $\mathcal{X} \subset \mathbb{R}^n$ and the function $F(x) = f(x) + l(x)$ where $f : \mathbb{R}^n \to \mathbb{R}$ is continuously differentiable and $l : \mathbb{R}^n \to \mathbb{R} \cup \{\infty\}$ is a*

*convex and possibly non-differentiable extended real-valued function. Let $x^* \in \mathcal{X}$ be the minimiser of $F$. We say that $F$ is proximal $\gamma$-quasar-convex with respect to $x^*$ if there exists $\mathcal{A} \subset \mathbb{R}_{>0}$ such that*

$$F(x^*) - F(x) \geq \frac{1}{\gamma} \langle Q_l(x, a), x^* - x \rangle, \qquad \forall \ x \in \mathcal{X}, \quad \forall \ a \in \mathcal{A}. \tag{8}$$

*Moreover, $F$ is proximal $\beta$-strongly-$\gamma$-quasar-convex with respect to $x^*$ if there exists $\mathcal{A} \subset \mathbb{R}_{>0}$ such that*

$$F(x^*) - F(x) \geq \frac{1}{\gamma} \langle Q_l(x, a), x^* - x \rangle + \frac{\beta}{2} \|x - x^*\|^2 \qquad \forall \ x \in \mathcal{X}, \quad \forall \ a \in \mathcal{A}. \tag{9}$$

As discussed in Remark 5 in Appendix A, when $l \equiv 0$, the proximal quasar-convexity reduces to quasar-convexity. Indeed, it shares similar properties to quasar-convexity. For example, any fixed point of a proximal quasar-convex function (i.e., $x = \text{Prox}_l(x - a\nabla f(x)))$ is a global minimum, and the minimiser of a proximal strongly-quasar-convex function is unique. Moreover, every proximal strongly-quasar-convex function satisfies the proximal error bound condition; see Appendix D. An example of a proximal quasar-convex function is $f(x, y) = x^2 y^2$ over $\mathcal{X} = \{(x, y) : x \geq 1\}$ or $\mathcal{X} = \{(x, y) : y \geq 1\}$, where $f$ is quasar-convex without constraints (for $\gamma \in (0, 4]$) and satisfies proximal quasar-convexity with the aforementioned constraints (with $l(\cdot) = \text{Proj}_{\mathcal{X}}(\cdot), \gamma \in (0, 2]$, and $a > 0$). Similarly, one can consider the function $f(x, y) = xy$. While this function is not quasar-convex, it is a proximal quasar-convex function over the constraint set $\mathcal{X} = \{(x, y) : x \geq 0, y \geq 0\}$ (with $l(\cdot) = \text{Proj}_{\mathcal{X}}(\cdot), \gamma \in (0, 1]$, and $a \in (0, 1]$).

Having this set up, the constrained Problem (1) can be equivalently written as

$$\min_x \{F(x) := f(x) + \mathcal{I}_{\mathcal{X}}(x)\}, \tag{10}$$

where $\mathcal{I}_{\mathcal{X}} : \mathbb{R}^n \to \mathbb{R} \cup \{\infty\}$ is the indicator function of the set $\mathcal{X}$; i.e.,

$$\mathcal{I}_{\mathcal{X}}(x) = \begin{cases} 0 & x \in \mathcal{X} \\ \infty & x \notin \mathcal{X}. \end{cases}$$

Therefore, for a constrained problem, Definition 2 is equivalent to

$$f(x^*) - f(x) \geq \frac{1}{\gamma} \langle P_{\mathcal{X}}(x, a), x^* - x \rangle + \frac{\beta}{2} \|x - x^*\|^2,$$

for any $x \in \mathcal{X}$, where

$$P_{\mathcal{X}}(x, a) \stackrel{\text{def}}{=} \frac{1}{a} [x - \text{Proj}_{\mathcal{X}}(x - a\nabla f(x))]. \tag{11}$$

is a special case of the function (6) with the proximal operator replaced by the projection operator and the function $l$ replaced by the indicator function corresponding to the set $\mathcal{X}$. The mapping $P_{\mathcal{X}}(x, a)$ can be viewed as a projected-gradient surrogate that plays the role of $\nabla f(x)$ in our proximal quasar-convex analysis.

In addition to quasar-convexity, we make a standard Lipschitz assumption on the gradient of the function $f$ for the derivation of the main results in the following section.

**Assumption 1** (Lipschitz Gradients). *Let $f : \mathbb{R}^n \to \mathbb{R}$ be a continuously differentiable function. Then there exists a constant $L_1 > 0$ such that*

$$\|\nabla f(x) - \nabla f(y)\| \leq L_1 \|x - y\|, \qquad \forall \ x, y \in \mathbb{R}^n, \tag{12}$$

*i.e., the gradient of $f$ is globally Lipschitz with constant $L_1$.*

Here, the index $L_1$ indicates that the Lipschitz constant corresponds to the Lipschitz constant of the derivative of $f$ and not the Lipschitz continuity of the function $f$ itself. While differentiability of $f$ is not needed for the implementation of Algorithm 1, Assumption 1 is a common assumption in the ZO optimisation literature (Nesterov & Spokoiny, 2017; Wang et al., 2020; Liu et al., 2018).

# 3 Convergence Properties of Algorithm 1

In this section, we present the main results of this paper in terms of convergence and iteration complexity of Algorithm 1 when applied to quasar-convex functions. The proofs of lemmas and theorems can be found in Appendix A and Appendix B, respectively. At a high level, we show that under (proximal) quasar-convexity the Gaussian-smoothing Algorithm 1 enjoys iteration complexity bounds that match those of first-order methods for convex and strongly convex problems, up to constants. The results are divided into the unconstrained setting (discussed in Section 3.1) and the constrained setting (discussed in Section 3.2).

## 3.1 Unconstrained Problem

In this section, we explore problem (1) with $\mathcal{X} = \mathbb{R}^n$. The following lemma characterises the behaviour of $f_\mu$ defined in (2), when $f$ is a quasar-convex function.

**Lemma 1.** *Let function $f : \mathbb{R}^n \to \mathbb{R}$ satisfy Assumption 1 and let $f$ be a $\gamma$-quasar-convex function with respect to some minimiser $x^* \in \mathbb{R}^n$. Then $f_\mu$ defined in (2) satisfies the inequality,*

$$\gamma(f_\mu(x^*) - f_\mu(x)) + \mu^2 L_1 n \geq \langle \nabla f_\mu(x), x^* - x \rangle, \qquad \forall \, x \in \mathbb{R}^n.$$

The proof of Lemma 1 can be found in Appendix A.

Now, using Lemma 1, we can build the bridge between quasar-convexity of $f$ and convergence of Algorithm 1. The following theorem and corollary characterise the convergence of Algorithm 1 when the objective function is quasar-convex.

**Theorem 1.** *Let $f : \mathbb{R}^n \to \mathbb{R}$ be $\gamma$-quasar-convex with respect to some minimiser $x^* \in \mathbb{R}^n$ and let $f$ satisfy Assumption 1. Consider the sequence $\{x_k\}_{k \geq 0}$ generated by Algorithm 1 with $t = 1$, step size $h_k = h = \frac{\gamma}{4(n+4)L_1}$ and let $\mathcal{U}_{k-1} = \{u_0, u_1, \ldots, u_{k-1}\}$. Then, for any $N \geq 0$, it holds that*

$$\frac{1}{N+1} \sum_{k=0}^{N} E_{\mathcal{U}_{k-1}}[f(x_k)] - f(x^*) \leq \frac{4(n+4)L_1}{\gamma^2} \frac{\|x_0 - x^*\|^2}{N+1} + \frac{2\mu^2 L_1 n(1+\gamma)}{\gamma} + \frac{\mu^2(n+6)^3 L_1}{8(n+4)}. \tag{13}$$

The proof of Theorem 1 can be found in Appendix B. From the upper bound in Theorem 1, the first right-hand side term of (13) is due to the initialisation error and becomes arbitrarily small for $N \to \infty$. The second and third terms are due to the error caused by the difference between the true function and the smoothed function, and using the random oracle defined in (3) instead of gradient of the function. These terms become arbitrarily small for $\mu \to 0$. The next corollary gives a guideline on how to choose the number of iterations and the smoothing parameter $\mu$ for a given specific tolerance $\epsilon$.

**Corollary 1.** *Adopt the hypothesis of Theorem 1. For a given tolerance $\epsilon > 0$, let $R = \|x_0 - x^*\|$. If*

$$\mu \leq \sqrt{\epsilon \left( \frac{4L_1 n(1+\gamma)}{\gamma} + \frac{(n+6)^3}{4(n+4)} L_1 \right)^{-1}} \quad and \quad N \geq \left\lceil \frac{8(n+4)L_1 R^2}{\gamma^2 \epsilon} - 1 \right\rceil,$$

*then,*

$$E_{\mathcal{U}_{N-1}}[f(\hat{x}_N)] - f(x^*) \leq \frac{1}{N+1} \sum_{k=0}^{N} E_{\mathcal{U}_{k-1}}[f(x_k)] - f(x^*) \leq \epsilon,$$

*where $\hat{x}_N = \arg\min_{x \in \{x_0, \ldots, x_N\}} f(x)$.*

The proof of Corollary 1 can be found in Appendix B. Corollary 1 implies that, in the quasar-convex setting, Algorithm 1 reaches an $\epsilon$-optimal point in $\mathcal{O}(n\epsilon^{-1})$ iterations, matching the dimension and accuracy dependence known for Gaussian-smoothing ZO methods on convex objectives.

**Remark 3.** *Considering a $\gamma$-quasar convex function, it is shown that gradient descent algorithm reaches an $\epsilon$-optimal point in $\mathcal{O}(\frac{R^2 L_1}{\gamma \epsilon})$ iterations (Jin, 2020), whereas our method (with $t = 1$) finds an $\epsilon$-optimal point*

in $\mathcal{O}(\frac{nR^2L_1}{\gamma^2\epsilon})$ iterations (see Corollary 1). Both bounds depend inversely on the step size, thus our choice of step size $h = \frac{\gamma}{4(n+4)L_1}$ yields an iteration complexity that has a higher order of inverse dependence on $\gamma$. we choose $h = \frac{1}{4(n+4)L_1}$ instead, we will have the iteration complexity of $\mathcal{O}(\frac{nR^2L_1}{\gamma\epsilon})$ but the choice of $\mu$ given in Corollary 1 will have an extra dependency on $\gamma$.

The following lemma characterises the behaviour of $f_\mu$ defined in (2), when $f$ is a strongly quasar convex function.

**Lemma 2.** *Let function $f : \mathbb{R}^n \to \mathbb{R}$ satisfy Assumption 1 and be a $\beta$-strongly-$\gamma$-quasar-convex function with respect to some minimiser $x^* \in \mathbb{R}^n$. Then $f_\mu$ defined in (2) satisfies the inequality*

$$\gamma(f_\mu(x^*) - f_\mu(x)) + \mu^2 L_1 n - \frac{\beta\gamma}{2}\|x - x^*\|^2 \geq \langle \nabla f_\mu(x), x^* - x \rangle.$$

The proof of Lemma 2 can be found in Appendix A. The additional negative quadratic term in Lemma 2 reflects the stronger curvature implied by strong quasar-convexity. Now, using Lemma 2, we characterise the convergence of Algorithm 1 when the objective function is strongly quasar-convex.

**Theorem 2.** *Let $f : \mathbb{R}^n \to \mathbb{R}$ be $\beta$-strongly-$\gamma$-quasar-convex with respect to some minimiser $x^* \in \mathbb{R}^n$ and satisfy Assumption 1. Consider the sequence $\{x_k\}_{k\geq 0}$ generated by Algorithm 1 with step size $h_k = h < \min\{\frac{\gamma}{2(n+4)L_1}, \frac{1}{\gamma\beta}\}$. Then, for any $N \geq 0$, we have*

$$\sum_{k=0}^{N}(1 - \gamma\beta h)^{N-k}(E_{\mathcal{U}_{k-1}}[f(x_k)] - f(x^*)) \leq \frac{(1-\gamma\beta h)^{N+1}R^2}{2h(\gamma - 2(n+4)L_1h)} + \frac{\mu^2 L_1 n(1+\gamma)}{h\gamma\beta(\gamma - 2(n+4)L_1h)}$$
$$+ \frac{\mu^2 L_1^2(n+6)^3}{4\gamma\beta(\gamma - 2(n+4)L_1h)}, \tag{14}$$

*where $\mathcal{U}_{k-1} = \{u_0, u_1, \ldots, u_{k-1}\}$, $R = \|x_0 - x^*\|$.*

The proof of Theorem 2 can be found in Appendix B. Given the upper bound provided by Theorem 2, the first right-hand side term of (14) is due to initialisation error and becomes arbitrarily small for $N \to \infty$. The second and third terms are due to the error caused by the difference between the true function and the smoothed function, and using the random oracle defined in (3) instead of gradient of the function. They become arbitrarily small for $\mu \to 0$. The next corollary gives a guideline on how to choose the number of iterations and the smoothing parameter $\mu$ for a given specific tolerance $\epsilon$.

**Corollary 2.** *Adopt the hypothesis of Theorem 2. For a given tolerance $\epsilon > 0$, let $a = \frac{L_1 n(1+\gamma)}{h\gamma\beta(\gamma-2(n+4)L_1h)}$, $b = \frac{L_1^2(n+6)^3}{4\gamma\beta(\gamma-2(n+4)L_1h)}$, and $q = 2h(\gamma - 2(n+4)L_1h)$. If*

$$\mu \leq \sqrt{\frac{\epsilon}{2}(a+b)^{-1}} \quad and \quad N \geq \left\lceil \frac{1}{\gamma\beta h}\log\left(\frac{R^2}{q\epsilon}\right) - 1 \right\rceil,$$

*then,*

$$E_{\mathcal{U}_{N-1}}[f(x_N)] - f(x^*) \leq \epsilon.$$

**Corollary 3.** *Adopting the hypothesis of Theorem 2, for a given tolerance $\epsilon > 0$, let $a = \frac{8L_1^2 n(1+\gamma)}{h\gamma^3\beta^3(\gamma-2(n+4)L_1h)}$, $b = \frac{2L_1^3(n+6)^3}{\gamma^3\beta^3(\gamma-2(n+4)L_1h)}$, and $q = 2h\gamma^2\beta^2(\gamma - 2(n+4)L_1h)$. If*

$$\mu \leq \sqrt{\frac{\epsilon}{2}(a+b)^{-1}} \quad and \quad N \geq \left\lceil \frac{1}{\gamma\beta h}\log\left(\frac{8L_1 R^2}{q\epsilon}\right) - 1 \right\rceil,$$

*then, $E_{\mathcal{U}_{N-1}}[\|x_N - x^*\|^2] \leq \epsilon$.*

The proof of Corollaries 2 and 3 can be found in Appendix B. Together, Theorems 1 and 2 and Corollaries 1–3 show that Algorithm 1 achieves $\mathcal{O}(n\epsilon^{-1})$ complexity in the quasar-convex case and $\mathcal{O}(n\log(\epsilon^{-1}))$ complexity in the strongly quasar-convex case, i.e., the same iteration-order as in the convex and strongly convex settings, despite the underlying objectives being nonconvex.

**Remark 4.** *Considering a $\beta$-strongly-$\gamma$-quasar-convex function, Hinder et al. (2020) showed that their proposed first-order algorithm can find an $\epsilon$-optimal point in $\mathcal{O}\left(\sqrt{\frac{L_1}{\beta\gamma^2}}\log\left(\frac{L_1}{\beta\gamma}\right)\log(\frac{1}{\gamma\epsilon})\right)$ iterations, whereas our method (with $t = 1$) finds an $\epsilon$-optimal point in $\mathcal{O}(\max\{1, \frac{nL_1}{\gamma^2\beta}\}\log(\frac{1}{q\epsilon}))$ iterations with $q$ given in Corollary 2. The complexity difference can be attributed to the choice of the step size and the limited information available in the ZO setting.*

## 3.2 Constrained Problem

In this subsection, we consider the constrained optimisation problem, where $\mathcal{X} \subset \mathbb{R}^n$ is a compact convex set. In the following, we make an assumption on the variance of the random ZO oracle.

**Assumption 2.** *The variance of oracle $g_\mu$ defined in (3) is upper bounded by $\sigma^2 \geq 0$; i.e.,*

$$E_u[||g_\mu(x) - \nabla f_\mu(x)||^2] \leq \sigma^2.$$

This is a common assumption in the literature of ZO and stochastic optimisation; see, for example, Maass et al. (2021); Liu et al. (2020a); Farzin et al. (2025a). The variance upper bound $\sigma^2$ can be approximated via $E_u[||g_\mu(x)||^2]$; see more details in Remark 6 in Appendix A. The variance $\sigma^2$ can be reduced through the simple variance reduction mechanism built into Algorithm 1. In each iteration, instead of sampling one direction $u$ and computing a single $g_\mu(x_k)$, we sample $t$ directions and use the average of the corresponding random oracles in the update step. Such mini-batch averaging is standard in ZO methods (Balasubramanian & Ghadimi, 2022) and leads to the variance scaling $E_u[||g_\mu(x_k) - \nabla f_\mu(x_k)||^2] \leq \sigma^2/t$ while preserving unbiasedness, i.e., $E_u[g_\mu(x_k)] = \nabla f_\mu(x_k)$. Now, we can characterise the convergence and iteration complexity of Algorithm 1 when the function is proximal quasar-convex.

**Theorem 3.** *Let $F : \mathbb{R}^n \to \mathbb{R} \cup \{\infty\}$ be the function defined in (10) and $\mathcal{X} \in \mathbb{R}^n$ be the constraint set defined in (1). Suppose that $F$ is proximal $\gamma$-quasar-convex with respect to $\mathcal{X}$ and some minimiser $x^* \in \mathcal{X}$ with $\mathcal{A}$ defined in Definition 2 and $f : \mathbb{R}^n \to \mathbb{R}$ satisfies Assumption 1. Assume that the constraint set $\mathcal{X}$ is compact and convex with $d_\mathcal{X}$ as its diameter. Consider the sequence $\{x_k\}_{k\geq 0}$ generated by Algorithm 1 with step size $h_k = h = \frac{\gamma}{4(n+4)L_1}$. Suppose that $h \in \mathcal{A}$. Then, under Assumption 2, for any $N \geq 0$, we have*

$$\frac{1}{N+1}\sum_{k=0}^{N} E_{\mathcal{U}_{k-1}}[F(x_k)] - F(x^*) \leq \frac{4(n+4)L_1}{\gamma^2}\frac{||x_0 - x^*||^2}{N+1} + \frac{d_\mathcal{X}\mu L_1(n+3)^{3/2}}{\gamma} + \frac{\mu^2(n+6)^3}{8(n+4)}L_1 + \frac{2d_\mathcal{X}\sigma}{\sqrt{t}\gamma}, \tag{15}$$

*where $\mathcal{U}_{k-1} = \{u_0, u_1, \ldots, u_{k-1}\}$.*

The proof of Theorem 3 can be found in Appendix B. The first term on the right-hand side of (15) is an initialisation term that decays as $\mathcal{O}(N^{-1})$, as in the unconstrained case. The second and third terms quantify the bias introduced by smoothing and by using the ZO oracle. These terms can be made small by choosing $\mu$ appropriately. The last term captures the effect of stochastic variance and decays as $\mathcal{O}(1/\sqrt{t})$; it can therefore be controlled by increasing the mini-batch size. Compared with Theorem 1, the constrained setting introduces precisely this additional variance term due to the projection step and the nonsmoothness. The next corollary provides a guideline on choosing the hyperparameters for a given specific tolerance $\epsilon > 0$.

**Corollary 4.** *Adopt the hypothesis of Theorem 3. For a given tolerance $\epsilon > 0$, let $R = ||x_0 - x^*||$, $a = \frac{(n+6)^3}{8(n+4)}L_1$, and $b = \frac{d_\mathcal{X}L_1(n+3)^{3/2}}{\gamma}$. If*

$$\mu \leq \frac{\sqrt{b^2 + 2a\epsilon} - b}{2a}, \quad t \geq \left\lceil\frac{64d_\mathcal{X}^2\sigma^2}{\epsilon^2\gamma^2}\right\rceil \quad and \quad N \geq \left\lceil\frac{16(n+4)L_1R^2}{\gamma^2\epsilon} - 1\right\rceil,$$

*then,*

$$E_{\mathcal{U}_{N-1}}(f(\hat{x}_N) - f(x^*)) \leq \frac{1}{N+1}\sum_{k=0}^{N} E_{\mathcal{U}_{k-1}}[f(x_k)] - f(x^*) \leq \epsilon$$

*where $\hat{x}_N = \arg\min_{x \in \{x_0, \ldots, x_N\}} f(x)$.*

The proof of Corollary 4 can be found in Appendix B. As can be seen, with appropriate choices of $(N, \mu, t)$, the projected Gaussian-smoothing ZO method is guaranteed to converge to an $\epsilon$-neighbourhood of the optimal value for proximal quasar-convex problems. This behaviour is consistent with other ZO methods on constrained nonconvex problems (Ghadimi et al., 2016; Liu et al., 2018), where only convergence to a neighbourhood of the optimum can be ensured in general. Nevertheless, the neighbourhood can be ensured to be arbitrarily small through the variance reduction step.

Next, we present the convergence of Algorithm 1 when the $f$ is proximal strongly-quasar-convex.

**Theorem 4.** *Let $F : \mathbb{R}^n \to \mathbb{R} \cup \{\infty\}$ be the function defined in (10) and $\mathcal{X}$ be the constraint set defined in (1). Suppose that $F$ is proximal $\beta$-strongly-$\gamma$-quasar-convex with respect to $\mathcal{X}$ and some minimiser $x^* \in \mathcal{X}$ with $\mathcal{A}$ defined in Definition 2 and $f : \mathbb{R}^n \to \mathbb{R}$ satisfies Assumption 1. Assume that the constraint set $\mathcal{X}$ is a compact convex set with $d_{\mathcal{X}}$ as its diameter. Consider the sequence $\{x_k\}_{k \geq 0}$ be generated by Algorithm 1 with step size $h_k = h \leq \min\{\frac{\gamma}{2(n+4)L_1}, \frac{1}{\gamma\beta}\}$. Suppose that $h \in \mathcal{A}$. Then, under Assumption 2, for any $N \geq 0$, we have*

$$\sum_{k=0}^{N}(1-\gamma\beta h)^{N-k}(E_{\mathcal{U}_{k-1}}[F(x_k)] - F(x^*)) \leq \frac{(1-\gamma\beta h)^{N+1}||x_0 - x^*||^2}{2h(\gamma - 2(n+4)L_1h)} + \frac{d_{\mathcal{X}}\sigma}{h\sqrt{t}\gamma\beta(\gamma - 2(n+4)L_1h)}$$
$$+ \frac{\mu^2 L_1^2(n+6)^3}{4\gamma\beta(\gamma - 2(n+4)L_1h)} + \frac{d_{\mathcal{X}}\mu L_1(n+3)^{3/2}}{2\gamma\beta h(\gamma - 2(n+4)L_1h)}, \quad (16)$$

*where $\mathcal{U}_{k-1} = \{u_0, u_1, \ldots, u_{k-1}\}$.*

The proof of Theorem 4 can be found in Appendix B. Given the upper bound provided by Theorem 4, the first right-hand side term of (16) becomes arbitrarily small for $N \to \infty$. The third and fourth terms, in turns, become arbitrarily small for $\mu \to 0$. The second term can be made arbitrarily small using the variance reduction technique. Comparing the bounds in Theorems 4 and 2, besides the terms caused by using the random oracle $g_\mu$ instead of the gradient and initialisation error, in the constrained case, there exists an extra term, caused by the variance of the random oracle. The next corollary gives a guideline on how to choose the hyperparameters for a given specific tolerance $\epsilon$.

**Corollary 5.** *Adopt the hypothesis of Theorem 4. For a given tolerance $\epsilon > 0$, let $R = \|x_0 - x^*\|$, $a = \frac{L_1^2(n+6)^3}{4\gamma\beta(\gamma-2(n+4)L_1h)}$, $b = \frac{d_{\mathcal{X}}L_1(n+3)^{3/2}}{2\gamma\beta h(\gamma-2(n+4)L_1h)}$, and $q = 2h(\gamma - 2(n+4)L_1h)$. If*

$$\mu \leq \frac{-b + \sqrt{b^2 + 4a\epsilon}}{2a}, \qquad t \geq \left\lceil \frac{4d_{\mathcal{X}}^2\sigma^2}{q^2\gamma^2\beta^2\epsilon^2} \right\rceil, \qquad and \qquad N \geq \left\lceil \frac{1}{\gamma\beta h}\log\left(\frac{R^2}{q\epsilon}\right) - 1 \right\rceil,$$

*then,*

$$E_{\mathcal{U}_{N-1}}[f(x_N)] - f(x^*) \leq \epsilon.$$

The proof of Corollary 5 can be found in Appendix B. In particular, for proximal strongly quasar-convex objectives, Algorithm 1 attains an $\epsilon$-accurate solution in $\mathcal{O}(n\log(\epsilon^{-1}))$ iterations, with an $\epsilon$-sized neighbourhood whose radius can be reduced arbitrarily by increasing $t$ and by decreasing $\mu$.

## 4 Numerical Examples

In this section, we illustrate the performance of Algorithm 1 via numerical examples. In our experiments, we use the performance of gradient descent (GD) as a benchmark. Although GD acquires first-order information about the function and thus is generally considered superior to ZO methods, we see that in some of our examples, Algorithm 1 performs comparably to, or even better than, GD. We report here a subset of representative scenarios and defer further details and additional experiments to Appendix E. Unless otherwise stated, curves show the average objective value and one standard deviation over multiple runs. In this section, we illustrate the performance of Algorithm 1 on several optimisation problems where quasar-convexity or a related structure is known or strongly suggested to hold. In all experiments, we use gradient descent (GD) as a benchmark. Although GD has access to first-order information and is often assumed to be superior to ZO schemes, we observe that in several examples, RM outperforms GD.

### 4.1 Learning Linear Dynamical System

We first consider linear dynamical system identification (LDSI), a setting where the objective is known to be quasar-convex under suitable assumptions (Hardt et al., 2018). Suppose that observations $\{u_i, y_i\}_{i=1}^{T} \subset \mathbb{R} \times \mathbb{R}$ are generated by a linear time-invariant system

$$x_{i+1} = Ax_i + Bu_i, \qquad y_i = Cx_i + Du_i + \xi_i,$$

where $x_i \in \mathbb{R}^n$ is the hidden state and $A \in \mathbb{R}^{n \times n}$, $B \in \mathbb{R}^{n \times 1}$, $C \in \mathbb{R}^{1 \times n}$, $D \in \mathbb{R}$ are unknown.

**Experimental setup.** We generate the true parameters and inputs following Hardt et al. (2018) with $n = 20$ and time horizon $T = 500$. Following Hinder et al. (2020), we generate 100 sequences at the beginning and minimise

$$\frac{1}{|\mathcal{B}|} \sum_{(u,y) \in \mathcal{B}} \left( \frac{1}{T - T_1} \sum_{i=T_1}^{T} (y_t - \hat{y}_t)^2 \right),$$

where $\mathcal{B}$ is a batch of 100 sequences and $T_1 = T/4$. The model is

$$\hat{x}_{i+1} = \hat{A}\hat{x}_i + \hat{B}u_i, \qquad \hat{y}_i = \hat{C}\hat{x}_i + \hat{D}u_i, \qquad \hat{x}_0 = 0,$$

with parameters $(\hat{A}, \hat{B}, \hat{C}, \hat{D})$ to be learned. The initial point $(\hat{A}_0, \hat{C}_0, \hat{D}_0)$ is obtained by perturbing the true parameters $(A, C, D)$ while ensuring that the spectral radius of $\hat{A}_0$ remains less than 1. We assume $B$ is known and set $\hat{B} = B$. The noise term is $\xi_i \sim 10^{-2}\mathcal{N}(0, I_n)$, as in Fu et al. (2023). The quasar-convexity parameter $\gamma$ derived in Hardt et al. (2018) is difficult to compute precisely in practice, so we tune it numerically in the simulations (the same holds for $L_1(f)$). We set $N = 1000$, $h = 10^{-6}$, $t = 1$, and $\mu = 10^{-6}$ for RM. For GD, we also experiment with step sizes $h = 10^{-5}$ and $h = 10^{-6}$ for comparison. Moreover, we consider a system of coupled mass-spring-dampers and generate the true parameters accordingly. We generate 200 sequences with $n = 10$, and time horizon $T = 300$. The details of the dynamical system are given in Appendix E.1. We set $N = 1500$, $h = 10^{-3}$, $t = 1$, and $\mu = 10^{-7}$ for RM. For GD, we also experiment with step sizes $h = 10^{-3}$ and $h = 5 \times 10^{-3}$ for comparison. As $t = 1$, the number of gradient calls for GD with fixed step size is the same as the number of iterations. The number of function calls for RM with a fixed step size is twice the number of iterations. We consider both of the algorithms with Armijo line search (LS) step size selection Armijo (1966), which iteratively reduces the step size until a sufficient decrease condition for the objective function is satisfied. The maximum possible step size for the Armijo LS settings is $h_0 = 10^{-1}$. We learn the parameters of the system using RM and GD, and then generate a sequence using the same inputs to compare the sequences generated by the learned systems with the sequence corresponding to the true system. For the sake of comparison with other ZO methods, we consider the CMA-ES method, explained thoroughly in (Hansen, 2016). We set the initial step size $\sigma_0 = 5 \times 10^{-2}$ (as suggested by (Hansen, 2016, Appendix A)) and population size of 18 (as suggested by (Hansen, 2016, eq (48))) for CMA-ES.

**Results.** First, for the dynamical system with random parameters, Figure 1 shows the average objective value and standard deviation over 5 runs, versus iterations and CPU time, for both RM and GD. Consistent with the discussion in Appendix F, the ZO method seems to exhibit favourable convergence properties compared to GD in this setting, and can achieve similar or better optimisation performance despite using only function evaluations. We should note that there are cases where GD diverges, and the gradient magnitude increases dramatically when $h = 10^{-5}$, which are excluded from Figure 1. Second, for the real physics-based system, Figure 2 shows the loss function values versus the number of iterations and CPU time, while Figure 4 shows the sequence generated by the predicted dynamical systems using RM and GD with the same inputs, compared to the ground truth sequence. It can be seen that, in this example and for fixed step size cases, RM outperforms GD (even with a smaller step size) and can follow the true sequence closely. Also, for the Armijo adaptive step-size case, we can see that RM outperform GD. Figure 3 presents the average of loss function values versus CPU time for CMA-ES, GD with Armijo LS, and Algorithm 1 with Armijo LS over 5 runs. It can be seen that, in this example, RM outperform both of the other methods. More discussion on the results is given in Appendix E.1.

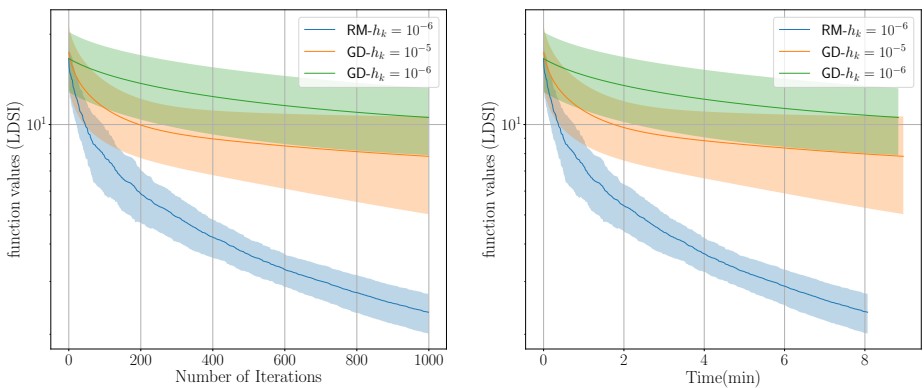

Figure 1: Learning linear dynamical systems with random parameters: average objective value (mean ± one standard deviation) over 4 runs, versus iterations (left) and CPU time (right), for RM and GD.

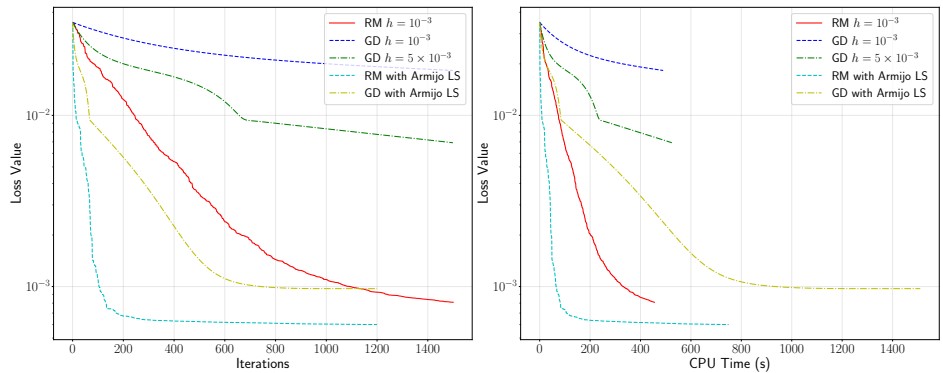

Figure 2: Learning real dynamical system: loss function values versus iterations (left) and CPU time (right).

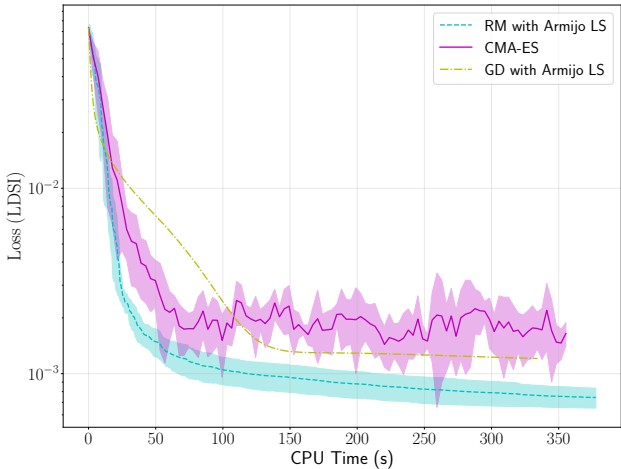

Figure 3: Learning real dynamical system: average of loss function values over 5 runs versus CPU time.

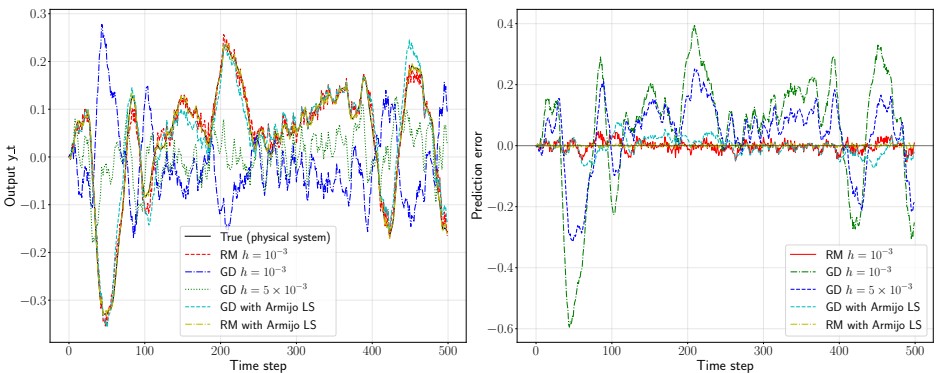

Figure 4: Generated sequences using the same input and predicted dynamical systems for RM and GD.

## 4.2 Black-Box Nonlinear System Identification

In the preceding experiments, gradients of the objective function were available and GD served as a benchmark. We now consider a setting where gradient information is *fundamentally unavailable*, making zeroth-order methods a necessity rather than a design choice.

We consider the problem of identifying the parameters of a dynamical system whose internal dynamics involve unknown non-linear components. Specifically, the true system is a physics-based simulator that, given an input sequence $\{u_i\}_{i=1}^T$ and candidate parameters $(\hat{A}, \hat{C}, \hat{D})$, returns an output sequence $\{\hat{y}_i\}_{i=1}^T$. The simulator contains actuator saturation and Coulomb friction, both of which are non-linear. Consequently, the loss function

$$\ell(\hat{A}, \hat{C}, \hat{D}) = \frac{1}{|\mathcal{B}|(T - T_1)} \sum_{(u,y) \in \mathcal{B}} \sum_{i=T_1}^{T} (\hat{y}_i - y_i)^2, \tag{17}$$

where $\hat{y}_i$ is obtained by rolling out the unknown nonlinear simulator with parameters $(\hat{A}, \hat{C}, \hat{D})$, cannot be differentiated with respect to the parameters. Gradient-based methods are therefore inapplicable, and zeroth-order optimisation is the only viable approach. This setting models practical scenarios such as interacting with compiled legacy simulation codes (Rios & Sahinidis, 2013; Audet & Hare, 2017), querying physical hardware through sensors and actuators (Maass et al., 2021), or interfacing with proprietary software whose source code is inaccessible.

We compare Algorithm 1 against CMA-ES (Hansen, 2016). Full details of the simulator, including the unknown nonlinear components and experimental parameters, are given in Appendix E.7. Figure 5 shows the average of the loss versus iterations and CPU time over 3 runs. It can be seen that Algorithm 1 with Armijo line search converges faster than the other algorithms.

## 4.3 Support Vector Machine with smoothed hinge loss function

We evaluate RM on a real-world classification task. We train a support vector machine (SVM) on the Breast Cancer dataset (Dua & Graff, 2019) using Algorithm 1. The SVM loss is the smoothed hinge loss from Hinder et al. (2020),

$$f(x) = \sum_{i=1}^{m} \phi_\alpha(1 - b_i a_i^T x),$$

where $a_i \in \mathbb{R}^n$, $b_i = \pm 1$ are given by the training data ($m = 569$, $n = 30$), and $\phi_\alpha(z) = 0$ for $z \le 0$, $\frac{z^2}{2}$ for $z \in [0, 1]$, and $\frac{z^\alpha - a}{\alpha} + \frac{1}{2}$ for $z \ge 1$. When $\alpha = 1$ the loss is convex, and for all $\alpha \in [0, 1]$ it is smooth and $\alpha$-quasar-convex. We choose $\alpha = 0.5$, $N = 10000$, $\mu = 10^{-7}$, $t = 1$, and five initial points sampled from $\mathcal{N}(0, I_n)$. As $t = 1$, the number of gradient calls for GD with fixed step size is the same as the number of iterations. The number of function calls for RM with a fixed step size is twice the number of iterations. We consider seven step sizes $[10^{-4}, 5 \times 10^{-5}, 10^{-5}, 5 \times 10^{-6}, 10^{-6}, 5 \times 10^{-7}, 10^{-7}]$. For each step size and initial

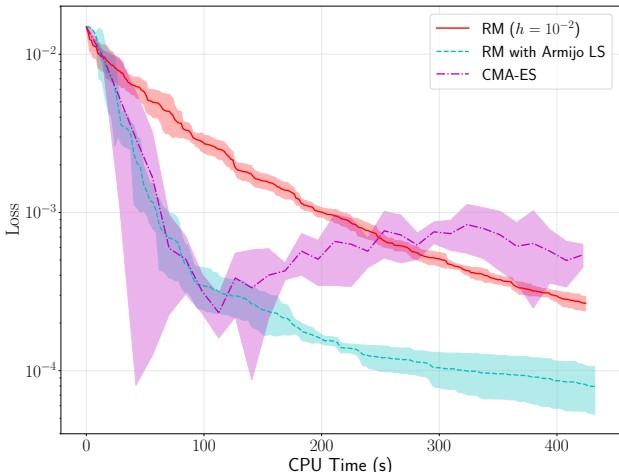

Figure 5: Black-box nonlinear system identification: average of loss function values versus CPU time for Algorithm 1 and CMA-ES over 3 runs.

point, RM is run four times and averaged. We run GD with the same step sizes for comparison. Table 1 reports the average CPU time (wall-clock time) and decay rate $1 - \frac{f(x_N)}{f(x_0)}$ over initialisations, for both RM and GD. Figure 6 shows the average loss and standard deviation versus iterations and CPU time. In this test, across all step sizes, RM closely tracks GD in both convergence speed and final objective value.

| Step size | RM | | GD | |
|---|---|---|---|---|
| | Time (s) | Decay Rate | Time (s) | Decay Rate |
| $1 \times 10^{-4}$ | 8.805 | 97.170 | 8.676 | 97.163 |
| $5 \times 10^{-5}$ | 8.925 | 96.693 | 8.708 | 96.714 |
| $1 \times 10^{-5}$ | 9.106 | 93.855 | 8.966 | 93.810 |
| $5 \times 10^{-6}$ | 9.112 | 91.841 | 9.210 | 91.839 |
| $1 \times 10^{-6}$ | 9.374 | 53.010 | 9.738 | 83.934 |
| $5 \times 10^{-7}$ | 9.540 | 65.256 | 10.190 | 74.523 |
| $1 \times 10^{-7}$ | 9.994 | 45.368 | 11.481 | 45.106 |

Table 1: Breast Cancer SVM with smoothed hinge loss: mean execution time and decay rate for RM and GD for different learning rates. In all scenarios, the two algorithms return almost identical solutions.

## 5 Conclusions And Future Research Directions

This paper analysed the performance of Gaussian-smoothing ZO random oracles for minimising (strong) quasar-convex functions, both with and without constraints. For the unconstrained problem, we established convergence and iteration complexity bounds of the ZO scheme when applied to quasar-convex and strongly quasar-convex loss functions, showing that it matches the order of guarantees known for convex and strongly convex objectives. For the constrained problem, we introduced the notion of proximal quasar-convexity and derived analogous complexity bounds for the projected ZO method, including the effect of variance reduction on the size of the attainable neighbourhood of the optimum. We also empirically illustrated regimes in which the ZO method behaves competitively with, and sometimes better than, gradient descent when learning a linear dynamical system, together with additional quasar-convex machine learning examples.

Several directions appear promising for future work. One is to extend the constrained analysis to settings with unbounded or more structured constraint sets that arise in control and reinforcement learning. A further future step is to analyse the convergence of Algorithm 1 with Armijo LS step size selection. Another is to

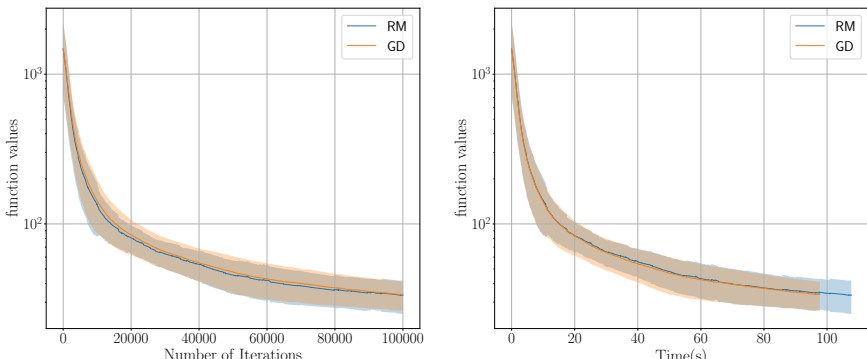

Figure 6: Breast Cancer SVM: average objective value (mean $\pm$ one standard deviation) for $h = 10^{-4}$, versus iterations (left) and CPU time (right), for RM and GD.

study min-max and saddle-point problems by exploiting quasar-convexity/concavity, for instance, in robust or adversarial training. A further avenue is to design adaptive ZO schemes that more directly leverage the averaged landscape interpretation of Gaussian smoothing in recurrent and deep network training.

## Acknowledgments

We sincerely thank the anonymous reviewers and the Action Editor for their constructive feedback and insights, which strengthened this manuscript. This work was supported by the Australian Research Council through a Discovery Project under Grant DP250101763 and the United States Air Force Office of Scientific Research under Grant No. FA2386-24-1-4014.

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

# A    Proofs of Auxiliary Results and background material

In this section, we collect proofs of auxiliary results and background material, which are necessary to show the main results of the paper. We start with proofs of Lemmas 1 and 2.

*Proof of Lemma 1.* Let $P(u) = e^{-\frac{1}{2}||u||^2}$. Then, from (2) we have

$$f_\mu(x) = \frac{1}{\kappa} \int_{\mathbb{R}^n} f(x + \mu u) P(u) du$$

$$\leq \frac{1}{\kappa} \int_{\mathbb{R}^n} \left( f(x^*) + \frac{1}{\gamma} \langle \nabla f(x + \mu u), x + \mu u - x^* \rangle \right) P(u) du$$

$$\leq \frac{1}{\kappa} \int_{\mathbb{R}^n} \left( f(x^* + \mu u) + \frac{1}{\gamma} \langle \nabla f(x + \mu u), x - x^* \rangle + \frac{1}{\gamma} \langle \nabla f(x + \mu u), \mu u \rangle \right) P(u) du$$

$$\leq f_\mu(x^*) + \frac{1}{\gamma} \langle \nabla f_\mu(x), x - x^* \rangle + \frac{\mu}{\gamma \kappa} \int_{\mathbb{R}^n} \langle \nabla f(x + \mu u), u \rangle P(u) du.$$

Here, the second line is due to the quasar-convexity of $f$ and the third line is due to the fact that $f(x^*) \leq f(x^* + \mu u)$ for all $u \in \mathbb{R}^n$. Thus we have

$$f_\mu(x^*) \geq f_\mu(x) + \frac{1}{\gamma} \langle \nabla f_\mu(x), x^* - x \rangle - \frac{\mu}{\gamma \kappa} \int \langle \nabla f(x + \mu u), u \rangle P(u) du,$$

or

$$\gamma(f_\mu(x^*) - f_\mu(x)) + E_u[\langle \nabla f(x + \mu u), \mu u \rangle] \geq \langle \nabla f_\mu(x), x^* - x \rangle. \tag{18}$$

Now, to bound $E_u[\langle \nabla f(x + \mu u), \mu u \rangle]$, we recall Assumption 1 and accordingly we have

$$f(x) \leq f(x + \mu u) + \langle \nabla f(x + \mu u), -\mu u \rangle + \frac{L_1 \mu^2}{2} \|u\|^2,$$

which is equivalent to

$$\langle \nabla f(x + \mu u), \mu u \rangle \leq f(x + \mu u) - f(x) + \frac{L_1 \mu^2}{2} \|u\|^2.$$

Computing the expectation with respect to $u$ and considering (Nesterov & Spokoiny, 2017, Lem. 1 & Thm. 1), we get

$$E_u[\langle \nabla f(x + \mu u), \mu u \rangle] \leq f_\mu(x) - f(x) + \frac{L_1 \mu^2 n}{2} \leq \mu^2 L_1 n. \tag{19}$$

(Here, the second inequality follows from (Nesterov & Spokoiny, 2017, Thm. 1).) Thus, combining (18) with (19) we obtain

$$\gamma(f_\mu(x^*) - f_\mu(x)) + \mu^2 L_1 n \geq \langle \nabla f_\mu(x), x^* - x \rangle,$$

which completes the proof. □

*Proof of Lemma 2.* As in the proof of Lemma 1, let $P(u) = e^{-\frac{1}{2}||u||^2}$. From (2) we have

$$f_\mu(x) = \frac{1}{\kappa} \int_{\mathbb{R}^n} f(x + \mu u) P(u) du$$

$$\leq \frac{1}{\kappa} \int_{\mathbb{R}^n} \left( f(x^*) + \frac{1}{\gamma} \langle \nabla f(x + \mu u), x + \mu u - x^* \rangle - \frac{\beta}{2} \|x + \mu u - x^*\|^2 \right) P(u) du$$

$$\leq \frac{1}{\kappa} \int_{\mathbb{R}^n} \left( f(x^* + \mu u) + \frac{1}{\gamma} \langle \nabla f(x + \mu u), x - x^* \rangle \right.$$

$$\left. + \frac{1}{\gamma} \langle \nabla f(x + \mu u), \mu u \rangle - \frac{\beta}{2} \|x - x^*\|^2 - \frac{\mu^2 \beta}{2} \|u\|^2 - 2\langle x - x^*, \mu u \rangle \right) P(u) du$$

$$\leq f_\mu(x^*) + \frac{1}{\gamma} \langle \nabla f_\mu(x), x - x^* \rangle - \frac{\beta}{2} \|x - x^*\|^2 + \frac{\mu}{\gamma \kappa} \int_{\mathbb{R}^n} \langle \nabla f(x + \mu u), u \rangle P(u) du.$$

The second line is due to the strong quasar-convexity of $f$. The third line is due to the fact that $f(x^*) \leq f(x^* + \mu u)$ for all $u$ since $x^*$ is a global minimiser. The last line is due to the fact that $E[\langle x - x^*, \mu u \rangle] = 0$. Moreover, from (Nesterov & Spokoiny, 2017, Lem. 1), $E[-\frac{\mu^2 \beta}{2}\|u\|^2] \leq -\frac{\mu^2 \beta n}{2}$ and a negative value can be eliminated from the upper bound. Thus we have

$$\gamma(f_\mu(x^*) - f_\mu(x)) + E_u[\langle \nabla f(x + \mu u), \mu u \rangle] - \frac{\beta \gamma}{2}\|x - x^*\|^2 \geq \langle \nabla f_\mu(x), x^* - x \rangle$$

Similar to the proof of Lemma 1, $E_u[\langle \nabla f(x + \mu u), \mu u \rangle]$ can be upper bounded, which completes the proof. □

**Remark 5.** *Consider an extension of the function $Q_l(x, a)$ in (6), i.e., consider*

$$Q_l(x, a) = \frac{1}{a}\Big(x - \arg\min_{z \in \mathcal{X}}\big[\|z - x\|^2 + l(z) - l(x) + 2a\langle \nabla f(x), z - x \rangle\big]\Big).$$

*We can see that*

$$\begin{aligned}
z' &= \arg\min_z \big[\|z - x\|^2 + 2a\langle \nabla f(x), z - x \rangle + l(z) - l(x)\big] \\
&= \arg\min_z \big[\frac{1}{a^2}\|z - x\|^2 + 2\langle \nabla f(x), z - x \rangle + \frac{1}{a^2}(l(z) - l(x))\big] \\
&= \arg\min_z \big[\frac{1}{a^2}\|z - x\|^2 + 2\langle \nabla f(x), z - x \rangle + \|\nabla f(x)\|^2 + \frac{1}{a^2}(l(z) - l(x))\big] \\
&= \arg\min_z \big[\|\frac{1}{a}(z - x) + \nabla f(x)\|^2 + \frac{1}{a^2}(l(z) - l(x))\big].
\end{aligned}$$

*Thus, if $l(\cdot)$ is constant, then $z' = x - a\nabla f(x)$, $Q_l(x, a) = \nabla f(x)$ and proximal $\gamma$-quasar-convexity reduces to $\gamma$-quasar-convexity independent of the positive scalar $a$.*

**Remark 6.** *If obtaining $\sigma$ is of interest, we can see that*

$$\begin{aligned}
E_u[\|g_\mu(x_k) - \nabla f_\mu(x_k)\|^2] &\leq E_u[\|g_\mu(x_k)\|^2 - 2\|g_\mu(x_k)\|\|\nabla f_\mu(x_k)\| + \|\nabla f_\mu(x_k)\|^2] \\
&\leq E_u[\|g_\mu(x_k)\|^2 - \|\nabla f_\mu(x_k)\|^2] \leq E_u[\|g_\mu(x_k)\|^2].
\end{aligned}$$

*The second inequality is due to $E_u[g_\mu(x_k)] = \nabla f_\mu(x_k)$. From this expression, an upper bound for $E_u[\|g_\mu(x_k)\|^2]$ can be obtained and the upper bounds can be candidates for $\sigma^2$. For example, from (Nesterov & Spokoiny, 2017, Thm. 4) we know that for a Lipschitz function $f$ we have $E_u[\|g_\mu(x)\|^2] \leq L_0(f)^2(n+4)^2$, where $L_0$ denotes the Lipschitz constant. Moreover, for a function $f$ with Lipschitz continuous gradients, we have $E_u[\|g_\mu(x)\|^2] \leq \frac{\mu^2}{2}L_1^2(f)(n+6)^3 + 2(n+4)\|\nabla f(x)\|^2$. These upper bounds can be used as candidates for $\sigma^2$.*

## B  Proofs of the theorems and corollaries

In this section, we give proofs of the main results presented in this paper.

*Proof of Theorem 1.* Let $r_k \stackrel{\text{def}}{=} \|x_k - x^*\|$, then we have

$$r_{k+1}^2 = \|x_k - hg_\mu(x_k) - x^*\|^2 \leq r_k^2 - 2h\langle g_\mu(x_k), x_k - x^* \rangle + h^2\|g_\mu(x_k)\|^2. \tag{20}$$

Taking the expectation with respect to $u_k$ and considering (Nesterov & Spokoiny, 2017, Them 4) leads to

$$E_{u_k}[r_{k+1}^2] \leq r_k^2 - 2h\langle \nabla f_\mu(x_k), x_k - x^* \rangle + h^2 \left( \frac{\mu^2(n+6)^3}{2} L_1^2 + 2(n+4)||\nabla f(x_k)||^2 \right)$$

$$\leq r_k^2 - 2h(\gamma(f_\mu(x) - f_\mu(x^*)) - \mu^2 L_1 n) + h^2(\frac{\mu^2(n+6)^3}{2} L_1^2 + 2(n+4)||\nabla f(x_k)||^2)$$

$$\leq r_k^2 - 2h(\gamma(f(x) - f(x^*) - \mu^2 L_1 n) - \mu^2 L_1 n) + h^2 \left[ \frac{\mu^2(n+6)^3}{2} L_1^2 + 4(n+4)L_1(f(x_k) - f(x^*)) \right]$$

$$\leq r_k^2 - 2h(\gamma - 2h(n+4)L_1)(f(x_k) - f(x^*)) + 2h\mu^2 L_1 n(1+\gamma) + \frac{\mu^2(n+6)^3}{2} L_1^2 h^2$$

$$= r_k^2 - h\gamma(f(x_k) - f(x^*)) + 2h\mu^2 L_1 n(1+\gamma) + \frac{\mu^2(n+6)^3}{2} L_1^2 h^2$$

The second inequality is due to Lemma 1 and the third one is due to (Nesterov & Spokoiny, 2017, Thm 1) and $f$ having Lipschitz gradients. Now, we take the expectations with respect to $\mathcal{U}_{k-1}$ and let $\rho_k \stackrel{\text{def}}{=} E_{\mathcal{U}_{k-1}}[r_k^2]$ and $\rho_0 = r_0^2$. Thus,

$$E_{\mathcal{U}_{k-1}}[f(x_k)] - f(x^*) \leq \frac{\rho_k - \rho_{k+1}}{\gamma h} + \frac{2\mu^2 L_1 n(1+\gamma)}{\gamma} + \frac{\mu^2(n+6)^3}{2\gamma} L_1^2 h. \qquad (21)$$

Summing this inequality from $k = 0$ to $k = N$ and dividing it by $N + 1$, yields

$$\frac{1}{N+1} \sum_{k=0}^{N} E_{\mathcal{U}_{k-1}}[f(x_k)] - f(x^*) \leq \frac{4(n+4)L_1}{\gamma^2} \frac{||x_0 - x^*||^2}{N+1} + \frac{2\mu^2 L_1 n(1+\gamma)}{\gamma} + \frac{\mu^2(n+6)^3}{8(n+4)} L_1,$$

which completes the proof. $\qquad \square$

*Proof of Corollary 1.* Adopting the hypothesis of Theorem 1, we want to upper bound the addition of side terms of (13) by $\epsilon$. Thus, by upper bounding each term dependent on $N$ and $\mu$ by $\frac{\epsilon}{2}$, we obtain the lower bound on the number of iterations $N$ and the upper bound on the smoothing parameter $\mu$. $\qquad \square$

*Proof of Theorem 2.* Let $r_k \stackrel{\text{def}}{=} ||x_k - x^*||$, then we have

$$r_{k+1}^2 = ||x_k - hg_\mu(x_k) - x^*||^2 \leq r_k^2 - 2h\langle g_\mu(x_k), x_k - x^* \rangle + h^2||g_\mu(x_k)||^2. \qquad (22)$$

Taking the expectation with respect to $u_k$ and considering (Nesterov & Spokoiny, 2017, Thm. 4) leads to

$$E_{u_k}[r_{k+1}^2] \leq r_k^2 - 2h\langle \nabla f_\mu(x_k), x_k - x^* \rangle + h^2(2(n+4)||\nabla f(x_k)||^2 + \frac{\mu^2}{2} L_1^2(n+6)^3)$$

$$\leq r_k^2 - 2h(\gamma(f_\mu(x_k) - f_\mu(x^*)) + \frac{\beta\gamma}{2} ||x_k - x^*||^2 - \mu^2 L_1 n)$$

$$+ h^2(4(n+4)L_1(f(x_k) - f(x^*)) + \frac{\mu^2}{2} L_1^2(n+6)^3)$$

$$\leq (1 - h\gamma\beta)r_k^2 - 2h(\gamma - 2(n+4)L_1 h)(f(x_k) - f(x^*)) + 2h\mu^2 L_1 n(1+\gamma) + \frac{h^2\mu^2 L_1^2(n+6)^3}{2}$$

$\qquad (23)$

The second inequality is due to Lemma 2 and $f_\mu$ having Lipschitz gradients. The third inequality is obtained by considering (Nesterov & Spokoiny, 2017, Thm. 1). Now, we take the expectations with respect to $\mathcal{U}_{k-1}$ and recursively apply the above inequality. Hence, we have that

$$\sum_{k=0}^{N} (1 - \gamma\beta h)^{N-k}(E_{\mathcal{U}_{k-1}}[f(x_k)] - f(x^*)) \leq \frac{(1 - \gamma\beta h)^{N+1}R^2}{2h(\gamma - 2(n+4)L_1 h)} + \frac{\mu^2 L_1 n(1+\gamma)}{h\gamma\beta(\gamma - 2(n+4)L_1 h)}$$

$$+ \frac{\mu^2 L_1^2(n+6)^3}{4\gamma\beta(\gamma - 2(n+4)L_1 h)}, \qquad (24)$$

which is obtained considering the geometric sequence summation rule

$$\sum_{k=0}^{N}(1-\gamma\beta h)^{N-k} = \sum_{k=0}^{N}\frac{(1-\gamma\beta h)^{N}}{(1-\gamma\beta h)^{k}} = \frac{1-(1-\gamma\beta h)^{N+1}}{(1-(1-\gamma\beta h))(1-\gamma\beta h)^{N}}(1-\gamma\beta h)^{N}$$
$$\leq \frac{1-(1-\gamma\beta h)^{N+1}}{1-(1-\gamma\beta h)} \leq \frac{1}{\gamma\beta h}, \tag{25}$$

which completes the proof. □

*Proof of Corollary 2.* Adopting the hypothesis of Theorem 2, we want to upper bound the addition of side terms of (14) by $\epsilon$. Thus, by upper bounding each term dependent on $N$ and $\mu$ by $\frac{\epsilon}{2}$, we obtain the lower bound on the number of iterations $N$ and the upper bound on the smoothing parameter $\mu$. By plugging in $a$, $b$, $q$, and $\mu$ into (14), we have

$$\sum_{k=0}^{N}(1-\gamma\beta h)^{N-k}(E_{\mathcal{U}_{k-1}}[f(x_k)] - f(x^*)) \leq \epsilon.$$

Since the left-hand side of the above inequality is the summation of $N+1$ positive terms, and each in expectation is less than $\epsilon$, we obtain the desired result. □

*Proof of Corollary 3.* Considering Lemma 3 in Appendix C, any strongly quasar-convex function with Lipschitz gradients satisfies the Polyak-Łojasiewicz (PL) inequality (see Definition 5), or equivalently, a quadratic growth condition (see Remark 10). Considering (14) and using Lemma 3 and Remark 10 in Appendix C, we have

$$\sum_{k=0}^{N}(1-\gamma\beta h)^{N-k}\frac{\gamma^2\beta^2}{8L_1}(E_{\mathcal{U}_{k-1}}[\|x_k - x^*\|^2]) \leq \frac{(1-\gamma\beta h)^{N+1}R^2}{2h(\gamma - 2(n+4)L_1 h)} + \frac{\mu^2 L_1 n(1+\gamma)}{h\gamma\beta(\gamma - 2(n+4)L_1 h)}$$
$$+ \frac{\mu^2 L_1^2(n+6)^3}{4\gamma\beta(\gamma - 2(n+4)L_1 h)}. \tag{26}$$

By plugging $a$, $b$, $q$ and $\mu$ into (26), we have

$$\sum_{k=0}^{N}(1-\gamma\beta h)^{N-k}E_{\mathcal{U}_{N-1}}[\|x_N - x^*\|^2] \leq \epsilon.$$

Since the left-hand side of the above inequality is the summation of $N+1$ positive terms and each in expectation is less than $\epsilon$, we obtain the desired result. □

Before we continue with a proof of Theorem 3, we define additional auxiliary variables. Considering Algorithm 1 and using (11), we define the variables

$$s_k \stackrel{\text{def}}{=} P_{\mathcal{X}}(x_k, g_\mu(x_k), h_k), \qquad v_k \stackrel{\text{def}}{=} P_{\mathcal{X}}(x_k, \nabla f_\mu(x_k), h_k), \qquad p_k \stackrel{\text{def}}{=} P_{\mathcal{X}}(x_k, \nabla f(x_k), h_k). \tag{27}$$

As a result, we can see that in the constrained case we have $x_{k+1} = x_k - h_k s_k$.

**Remark 7.** *Considering Algorithm 1 and Definition 2 with $l(x) = I_{\mathcal{X}}(x)$, we have $\text{Prox}_l(\cdot) = \text{Proj}_{\mathcal{X}}(\cdot)$, $Q_l(x, h_k) = p_k$ and $F(x^*) - F(x_k) \geq \frac{1}{\gamma}\langle Q_l(x_k, h_k), x^* - x_k\rangle$ or $f(x^*) - f(x_k) \geq \frac{1}{\gamma}\langle p_k, x^* - x_k\rangle$, which will be used in the proof of Theorem 3.*

*Proof of Theorem 3.* Let $r_k \overset{\text{def}}{=} \|x_k - x^*\|$ and $g_\mu(x_k) - \nabla f_\mu(x_k) = \xi_k$, and let the auxiliary variables $s_k$, $v_k$, $p_k$ be defined in (27). Then we have

$$
\begin{aligned}
r_{k+1}^2 &= \|x_k - hs_k - x^*\|^2 \\
&\le r_k^2 - 2h\langle s_k, x_k - x^*\rangle + h^2\|s_k\|^2 \\
&\le r_k^2 - 2h\langle p_k, x_k - x^*\rangle - 2h\langle s_k - v_k, x_k - x^*\rangle - 2h\langle v_k - p_k, x_k - x^*\rangle + h^2\|g_\mu(x_k)\|^2 \\
&\le r_k^2 - 2h\langle p_k, x_k - x^*\rangle + 2h\|s_k - v_k\|\|x_k - x^*\| + 2h\|v_k - p_k\|\|x_k - x^*\| + h^2\|g_\mu(x_k)\|^2 \\
&\le r_k^2 - 2h\langle p_k, x_k - x^*\rangle + 2hd_\mathcal{X}\|g_\mu(x_k) - \nabla f_\mu(x_k)\| + 2hd_\mathcal{X}\|\nabla f_\mu(x_k) - \nabla f(x_k)\| + h^2\|g_\mu(x_k)\|^2 \\
&\le r_k^2 - 2h\langle p_k, x_k - x^*\rangle + 2hd_\mathcal{X}\|\xi_k\| + hd_\mathcal{X}\mu L_1(n+3)^{3/2} + h^2\|g_\mu(x_k)\|^2.
\end{aligned}
\tag{28}
$$

The second inequality is due to $\|s_k\| \le \|g_\mu(x_k)\|$ which can be obtained from (Ghadimi et al., 2016, Lem. 1), noting that the function $h$ defined in (Ghadimi et al., 2016, (1)) is the constraint set indicator function with $h(x_k) = 0$ and $\alpha = 1$. The latter is the consequence of the fact that in our case $V(x,z)$ defined in (Ghadimi et al., 2016, (8)) is equal to $\|x - z\|_2^2/2$. The fourth inequality is due to $\|s_k - v_k\| \le \|g_\mu(x_k) - \nabla f_\mu(x_k)\|$ and $\|v_k - p + k\| \le \|\nabla f_\mu(x_k) - \nabla f(x_k)\|$, which can be obtained directly from (Ghadimi et al., 2016, Prop. 1) (letting $\alpha = 1$). The last inequality is due to (Nesterov & Spokoiny, 2017, Lem. 3). Taking the expectation with respect to $u_k$ and considering $I_\mathcal{X}(x^*) = I_\mathcal{X}(x_k) = 0$, $E_{u_k}[\|\xi_k\|] \le \frac{\sigma}{\sqrt{t}}$ (due to Jensen inequality $E_{u_k}[\|\xi_k\|]^2 \le E_{u_k}[\|\xi_k\|^2] \le \frac{\sigma^2}{t}$), (Nesterov & Spokoiny, 2017, Thm. 4), Remark 7, and Definition 2 lead to

$$
\begin{aligned}
E_{u_k}[r_{k+1}^2] &\le r_k^2 - 2h\langle p_k, x_k - x^*\rangle + 2hd_\mathcal{X}\frac{\sigma}{\sqrt{t}} + hd_\mathcal{X}\mu L_1(n+3)^{\frac{3}{2}} + h^2\left(\frac{\mu^2(n+6)^3}{2}L_1^2 + 2(n+4)\|\nabla f(x_k)\|^2\right) \\
&\le r_k^2 - 2h\gamma(F(x_k) - F(x^*)) + 2hd_\mathcal{X}\frac{\sigma}{\sqrt{t}} + hd_\mathcal{X}\mu L_1(n+3)^{3/2} + h^2\left(\frac{\mu^2(n+6)^3}{2}L_1^2 + 2(n+4)\|\nabla f(x_k)\|^2\right) \\
&\le r_k^2 - 2h\gamma(F(x_k) - F(x^*)) + 2hd_\mathcal{X}\frac{\sigma}{\sqrt{t}} + hd_\mathcal{X}\mu L_1(n+3)^{3/2} + h^2\left[\frac{\mu^2(n+6)^3}{2}L_1^2 + 4(n+4)L_1(f(x_k) - f(x^*))\right] \\
&\le r_k^2 - 2h(\gamma - 2h(n+4)L_1)(F(x_k) - F(x^*)) + 2hd_\mathcal{X}\frac{\sigma}{\sqrt{t}} + hd_\mathcal{X}\mu L_1(n+3)^{3/2} + \frac{\mu^2(n+6)^3}{2}L_1^2 h^2 \\
&= r_k^2 - h\gamma(F(x_k) - F(x^*)) + 2hd_\mathcal{X}\frac{\sigma}{\sqrt{t}} + hd_\mathcal{X}\mu L_1(n+3)^{3/2} + \frac{\mu^2(n+6)^3}{2}L_1^2 h^2
\end{aligned}
\tag{29}
$$

Now, we take the expectations with respect to $\mathcal{U}_{k-1}$ and let $\rho_k \overset{\text{def}}{=} E_{\mathcal{U}_{k-1}}[r_k^2]$ and $\rho_0 = r_0^2$. Thus,

$$
E_{\mathcal{U}_{k-1}}[F(x_k)] - F(x^*) \le \frac{\rho_k - \rho_{k+1}}{\gamma h} + \frac{d_\mathcal{X}\mu L_1(n+3)^{3/2}}{\gamma} + \frac{\mu^2(n+6)^3}{2\gamma}L_1^2 h + \frac{2d_\mathcal{X}\sigma}{\gamma\sqrt{t}}.
\tag{30}
$$

Summing this inequality from $k = 0$ to $k = N$ and dividing it by $N + 1$, yields

$$
\frac{1}{N+1}\sum_{k=0}^{N} E_{\mathcal{U}_{k-1}}[F(x_k)] - F(x^*) \le \frac{4(n+4)L_1}{\gamma^2}\frac{\|x_0 - x^*\|^2}{N+1} + \frac{d_\mathcal{X}\mu L_1(n+3)^{3/2}}{\gamma} + \frac{\mu^2(n+6)^3}{8(n+4)}L_1 + \frac{2d_\mathcal{X}\sigma}{\gamma\sqrt{t}}
$$

which completes the proof. $\square$

*Proof of Corollary 4.* Adopting the hypothesis of Theorem 3, we want to upper bound the right-hand side of (15) by $\epsilon$. Thus, by upper bounding each term dependent on $N$ and $t$ by $\frac{\epsilon}{4}$, and upper bounding the term dependent on $\mu$ by $\frac{\epsilon}{2}$, we obtain the bounds on $N$, $t$, and $\mu$. $\square$

*Proof of Theorem 4.* Let $r_k \stackrel{\text{def}}{=} ||x_k - x^*||$ and $\xi_k = g_\mu(x_k) - \nabla f_\mu(x_k)$, and let $s_k$, $v_k$, $p_k$ be defined in (27). Then

$$
\begin{aligned}
r_{k+1}^2 &= ||x_k - hs_k - x^*||^2 \\
&\leq r_k^2 - 2h\langle s_k, x_k - x^*\rangle + h^2||s_k||^2 \\
&\leq r_k^2 - 2h\langle p_k, x_k - x^*\rangle - 2h\langle s_k - v_k, x_k - x^*\rangle - 2h\langle v_k - p_k, x_k - x^*\rangle + h^2||g_\mu(x_k)||^2 \\
&\leq r_k^2 - 2h\langle p_k, x_k - x^*\rangle + 2h||s_k - v_k|| ||x - x^*|| + ||v_k - p_k|| ||x - x^*|| + h^2||g_\mu(x_k)||^2 \\
&\leq r_k^2 - 2h\langle p_k, x_k - x^*\rangle + 2hd_\mathcal{X}||g_\mu(x_k) - \nabla f_\mu(x_k)|| + 2hd_\mathcal{X}||\nabla f_\mu(x_k) - \nabla f(x_k)|| + h^2||g_\mu(x_k)||^2 \\
&\leq r_k^2 - 2h\langle p_k, x_k - x^*\rangle + 2hd_\mathcal{X}||\xi_k|| + hd_\mathcal{X}\mu L_1(n+3)^{3/2} + h^2||g_\mu(x_k)||^2.
\end{aligned}
\tag{31}
$$

The second inequality is due to $||s_k|| \leq ||g_\mu(x_k)||$ which can be obtained from (Ghadimi et al., 2016, Lem. 1) noting that the function $h$ defined in (Ghadimi et al., 2016, (1)) is the constraint set indicator function with $h(x_k) = 0$ and $\alpha = 1$. The latter is the consequence of the fact that in our case $V(x, z)$ defined in (Ghadimi et al., 2016, (8)) is equal to $||x - z||_2^2/2$. The fourth inequality is due to $||s_k - v_k|| \leq ||g_\mu(x_k) - \nabla f_\mu(x_k)||$ and $||v_k - p_k|| \leq ||\nabla f_\mu(x_k) - \nabla f(x_k)||$, which can be obtained directly from (Ghadimi et al., 2016, Prop. 1) (letting $\alpha = 1$). The last inequality is due to (Nesterov & Spokoiny, 2017, Lem. 3). Taking the expectation with respect to $u_k$ and considering $I_\mathcal{X}(x^*) = I_\mathcal{X}(x_k) = 0$, $E_{u_k}[||\xi_k||] \leq \frac{\sigma}{\sqrt{t}}$ (due to Jensen inequality $E_{u_k}[||\xi_k||]^2 \leq E_{u_k}[||\xi_k||^2] \leq \frac{\sigma^2}{t}$), (Nesterov & Spokoiny, 2017, Thm. 4), Remark 7, and Definition 2 lead to

$$
\begin{aligned}
E_{u_k}[r_{k+1}^2] &\leq r_k^2 - 2h\langle p_k, x_k - x^*\rangle + 2hd_\mathcal{X}\frac{\sigma}{\sqrt{t}} + hd_\mathcal{X}\mu L_1(n+3)^{\frac{3}{2}} + h^2(\frac{\mu^2}{2}L_1^2(n+6)^3 + 2(n+4)||\nabla f(x)||^2) \\
&\leq r_k^2 - 2h\gamma(F(x_k) - F(x^*) + \frac{\beta}{2}||x_k - x^*||^2) + 2hd_\mathcal{X}\frac{\sigma}{\sqrt{t}} + hd_\mathcal{X}\mu L_1(n+3)^{3/2} \\
&\quad + h^2(\frac{\mu^2}{2}L_1^2(n+6)^3 + 2(n+4)||\nabla f(x)||^2) \\
&\leq r_k^2 - 2h\gamma(F(x_k) - F(x^*) + \frac{\beta}{2}||x_k - x^*||^2) + 2hd_\mathcal{X}\frac{\sigma}{\sqrt{t}} + hd_\mathcal{X}\mu L_1(n+3)^{\frac{3}{2}} \\
&\quad + h^2(\frac{\mu^2}{2}L_1^2(n+6)^3 + 4(n+4)L_1(f(x_k) - f(x^*))) \\
&\leq (1 - h\gamma\beta)r_k^2 - 2h(\gamma - 2(n+4)L_1h)(F(x_k) - F(x^*)) + 2hd_\mathcal{X}\frac{\sigma}{\sqrt{t}} + hd_\mathcal{X}\mu L_1(n+3)^{3/2} + h^2\frac{\mu^2}{2}L_1^2(n+6)^3.
\end{aligned}
$$

We take the expectations with respect to $\mathcal{U}_{k-1}$ and recursively apply the above inequality, leading to

$$
\begin{aligned}
\sum_{k=0}^N (1 - \gamma\beta h)^{N-k}(E_{\mathcal{U}_{k-1}}[F(x_k)] - F(x^*)) &\leq \frac{(1 - \gamma\beta h)^{N+1}R^2}{2h(\gamma - 2(n+4)L_1h)} + \frac{d_\mathcal{X}\sigma}{h\sqrt{t}\gamma\beta(\gamma - 2(n+4)L_1h)} \\
&\quad + \frac{\mu^2 L_1^2(n+6)^3}{4\gamma\beta(\gamma - 2(n+4)L_1h)} + \frac{d_\mathcal{X}\mu L_1(n+3)^{3/2}}{2\gamma\beta h(\gamma - 2(n+4)L_1h)},
\end{aligned}
\tag{32}
$$

which is again obtained using the geometric sequence summation rule in (25).

$\square$

*Proof of Corollary 5.* Adopting the hypothesis of Theorem 4, we want to upper bound the addition of side terms of (16) by $\epsilon$. Thus, by upper bounding each term dependent on $N$ and $t$ by $\frac{\epsilon}{4}$, and by upper bounding the terms dependent on $\mu$ by $\frac{\epsilon}{2}$, we obtain the lower bound on the number of iterations $N$, $t$, and the upper bound on the smoothing parameter $\mu$. By plugging $a$, $b$, $q$, and $\mu$ into (16), we have

$$
\sum_{k=0}^N (1 - \gamma\beta h)^{N-k}(E_{\mathcal{U}_{k-1}}[f(x_k)] - f(x^*)) \leq \epsilon.
$$

Since the left-hand side of the above inequality is the summation of $N + 1$ positive terms and each in expectation is less than $\epsilon$, we obtain the desired result. $\square$

## C  Some Other Function Classes Related to Quasar-Convexity

In the introduction in Section 1, we pointed to convex, star-convex and Polyak-Łojasiewicz (PL) functions besides the class of (strong) quasar-convex functions. In this section, for the sake of completeness, we provide definitions of these function classes and highlight important connections and differences. For a thorough presentation on the topic with more details, we refer to Hinder et al. (2020). As an important result in the context of this paper, we prove that a PL condition can be implied by strong quasar-convexity and we give the relation between their corresponding parameters.

**Definition 3** ((Strong) Star-convexity). *Let $x^*$ be a minimiser of the differentiable function $f : \mathbb{R}^n \to \mathbb{R}$. The function $f$ is star-convex with respect to $x^*$ if for all $x \in \mathbb{R}^n$,*

$$f(x^*) \geq f(x) + \langle \nabla f(x), x^* - x \rangle,$$

*The function is strongly star-convex with respect to $x^*$ if for all $x \in \mathbb{R}^n$,*

$$f(x^*) \geq f(x) + \langle \nabla f(x), x^* - x \rangle + \frac{\beta}{2} \|x - x^*\|^2.$$

**Remark 8.** *(Strong) Star-convexity is a special case of (strong) quasar-convexity when $\gamma$ in Definition 1 is equal to 1.*

**Definition 4** ((Strong) Convexity). *Let the function $f : \mathbb{R}^n \to \mathbb{R}$ be differentiable. The function $f$ is convex if for all $x_1, x_2 \in \mathbb{R}^n$,*
$$f(x_2) \geq f(x_1) + \langle \nabla f(x_1), x_2 - x_1 \rangle.$$
*The function is strongly convex if for all $x_1, x_2 \in \mathbb{R}^n$,*

$$f(x_2) \geq f(x_1) + \langle \nabla f(x_1), x_2 - x_1 \rangle + \frac{\beta}{2} \|x_1 - x_2\|^2.$$

**Remark 9.** *(Strong) Convexity is a special case of (strong) star-convexity when $x_2$ in Definition 4 is fixed at the minimiser $x^*$.*

**Definition 5** (Polyak-Łojasiewicz functions). *Let $\eta > 0$ and $x^*$ be a minimiser of the differentiable function $f : \mathbb{R}^n \to \mathbb{R}$. The function $f$ is PL if for all $x \in \mathbb{R}^n$,*

$$\frac{1}{2} \|\nabla f(x)\|^2 \geq \eta(f(x) - f(x^*)) \tag{33}$$

**Lemma 3.** *Let $f : \mathbb{R}^n \to \mathbb{R}$ be a $\beta$-strongly-$\gamma$-quasar-convex function satisfying Assumption 1. Then $f$ satisfies the PL condition (33) with $\frac{\gamma^2 \beta^2}{4L_1}$ as the PL parameter $\eta$.*

*Proof.* Considering Definition 1 and the fact that $f(x^*) - f(x) \leq 0$, we have

$$\langle \nabla f(x), x - x^* \rangle \geq \frac{\gamma \beta}{2} \|x^* - x\|^2.$$

Considering the Cauchy-Schwarz inequality, we obtain

$$\|\nabla f(x)\| \geq \frac{\gamma \beta}{2} \|x^* - x\|.$$

From the Lipschitz gradient inequality (12) and the fact that $\nabla f(x^*) = 0$, we have

$$f(x) \leq f(x^*) + \langle \nabla f(x^*), x - x^* \rangle + \frac{L_1}{2} \|x - x^*\|^2 \leq f(x^*) + \frac{L_1}{2} \|x - x^*\|^2.$$

Thus

$$f(x) - f(x^*) \leq \frac{2L_1}{\gamma^2 \beta^2} \|\nabla f(x)\|^2, \qquad \text{or} \qquad \frac{1}{2} \|\nabla f(x)\|^2 \geq \frac{\gamma^2 \beta^2}{4L_1} (f(x) - f(x^*)).$$

$\square$

**Remark 10.** *From Karimi et al. (2016), it is known that if a function $f : \mathbb{R}^n \to \mathbb{R}$ satisfies the PL condition (33) with $\eta$ as the PL parameter, then $f$ satisfies the quadratic growth condition $f(x) - f(x^*) \geq \frac{\eta}{2}\|x - x^*\|^2$. Thus if $f$ is a $\beta$-strongly-$\gamma$-quasar-convex function with Lipschitz gradients, we have*

$$f(x) - f(x^*) \geq \frac{\gamma^2 \beta^2}{8L_1}\|x - x^*\|^2,$$

*which follows from Lemma 3.*

## D Strong Proximal Quasar-Convexity Implies Proximal Error Bound

It is known that strong quasar-convexity implies the PL condition in (33). One may wonder whether the proximal version of this condition has a similar relation. It turns out that a strong proximal quasar-convex function indeed satisfies a proximal error bound condition (or equivalently, proximal Polyak-Łojasiewicz or Kurdyka-Łojasiewicz condition) Karimi et al. (2016).

Suppose that $F = f + l$ is strongly proximal quasar-convex according to Definition 2. Using the definition of strong proximal quasar-convexity, we have

$$F(x^*) - F(x) \geq \frac{1}{a\gamma}\langle \mathrm{Prox}_{al}(x - a\nabla f(x)) - x, x^* - x \rangle + \frac{\beta}{2}\|x - x^*\|^2.$$

This implies that

$$\frac{1}{a\gamma}\langle x - \mathrm{Prox}_{al}(x - a\nabla f(x)), x^* - x \rangle + \frac{\beta}{2}\|x - x^*\|^2 \leq 0.$$

When $x = x^*$, the proximal error bound condition holds trivially. Now, let us assume that $x \neq x^*$. Rearranging terms, we have

$$\frac{\beta}{2}\|x^* - x\|^2 \leq \frac{1}{a\gamma}\langle \mathrm{Prox}_{al}(x - a\nabla f(x)) - x, x^* - x \rangle \leq \frac{1}{a\gamma}\|\mathrm{Prox}_{al}(x - a\nabla f(x)) - x\|\|x^* - x\|.$$

Therefore, if we let $\mathcal{X}^*$ denote the set of minimisers of $F$, we see that

$$\min_{x_p \in \mathcal{X}^*} \|x - x_p\| \leq \|x^* - x\| \leq \frac{2}{a\beta\gamma}\|\mathrm{Prox}_{al}(x - a\nabla f(x)) - x\|,$$

as desired.

## E Numerical Examples Complementary Details

In this section, we present additional numerical examples to illustrate the theoretical findings in Section 3. All experiments are implemented in Python and run on a Dell Latitude 7430 laptop with an Intel Core i7-1265U processor.

### E.1 Learning a Mechanical Linear Dynamical System

We consider a chain of 5 coupled mass-spring-damper oscillators ($n = 10$ states) illustrated in Figure 7 and discretised from the continuous-time model

$$\dot{x} = A_c x + B_c u, \qquad y = C_c x + D_c u. \tag{34}$$

Here, the state is defined as $x = [x_1, v_1, x_2, v_2, x_3, v_3, x_4, v_4, x_5, v_5]^\top$, where $x_i, v_i$ are position and velocity of mass $i$. The continuous-time dynamics for mass $i = 1, \ldots, 5$ ($m_i$, $k_i = 2.0$, $c_i = 0.4$, coupling $k_{\mathrm{cpl}} = 0.6$) are:

$$\dot{x}_i = v_i, \tag{35}$$

$$\dot{v}_i = \frac{1}{m_i}\Big[u - k_i x_i - c_i v_i + k_{\mathrm{cpl}}(x_{i-1} - x_i) + k_{\mathrm{cpl}}(x_{i+1} - x_i)\Big], \tag{36}$$

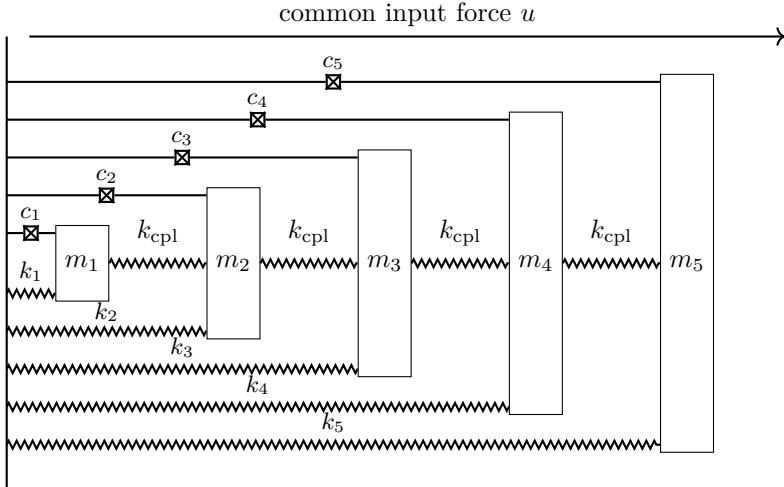

Figure 7: Chain of five coupled mass–spring–damper oscillators.

with $x_0 = x_6 = 0$ (fixed wall), $u$ (common input force), and masses $m = [1.0, 1.1, 1.2, 1.0, 0.9]$. Accordingly, the matrices in (34) are defined as

$$A_c = \begin{bmatrix} 0 & 1 & 0 & 0 & \cdots & 0 \\ a_{1,1} & b_1 & e_1 & 0 & \cdots & 0 \\ 0 & 0 & 0 & 1 & 0 & \cdots \\ e_2 & 0 & a_{2,2} & b_2 & e_2 & \cdots \\ \vdots & & & \ddots & \ddots & \ddots \end{bmatrix}, \quad B_c = \begin{bmatrix} 0 \\ 1/m_1 \\ 0 \\ 1/m_2 \\ \vdots \\ 1/m_5 \end{bmatrix}, \quad C_c = \begin{bmatrix} 0, 0, 0.2, 0, 1, 0, 0.2, 0, 0.2, 0 \end{bmatrix}, \quad D_c = 0,$$

(37)

where $a_{1,1} = -(k_1 + k_{\mathrm{cpl}})/m_1$, $a_{5,5} = -(k_5 + k_{\mathrm{cpl}})/m_5$, $a_{i,i} = -(k_i + 2k_{\mathrm{cpl}})/m_i$ for $i \in \{2, 3, 4\}$ and $b_i = -c_i/m_i$, $e_i = k_{\mathrm{cpl}}/m_i$ for $i \in \{1, \ldots, 5\}$. We discretise the continuous-time system using forward Euler with time step $T_s = 0.05$s, the discrete-time dynamics matrices are

$$A = I + T_s A_c, \quad B = T_s B_c, \quad C = C_c, \quad D = D_c,$$

(38)

and the spectral radius of $A$ satisfies $\|A\| < 1$, i.e., stability is preserved through the time step selection $T_s$. This model represents realistic wave propagation and vibration modes in a coupled mechanical chain. As explained in Section 4.1, we learn a dynamical system using RM and GD to be able to follow the outputs of the true dynamical system for the same inputs. We should note that the Armijo LS step size selection for RM does not use the gradient of the cost function and it uses the random oracle (3) instead. It can be seen in Figure 2 that for fixed step sizes, RM outperforms GD in minimising this loss function, even with a 5 times smaller step size. We can see the same for the case of the Armijo LS step size selection. It can be seen that the performance of GD with Armijo LS step size selection is comparable with RM with a fixed step size. In Figure 4, we compared the generated sequence by the systems learned by RM and GD for the same input with the true sequence. It can be seen that the sequence generated by RM, with both fixed and Armijo LS step sizes, closely follows the true sequence, while GD follows true sequence only with Armijo LS step size selection. We should note that the algorithms are learning a dynamical system to be able to follow the output of the true system with the same input distribution of the generated dataset, as desired. In Table 2, we can see that the mean squared error of the sequence generated by RM is 100 times smaller than that of GD in the fixed step size case, and it is 10 times smaller in the Armijo LS step size case.

Table 2: Test MSE for the coupled mass chain

|  | RM-fixed | RM-Armijo | GD-small | GD-large | GD-Armijo | Initial |
|---|---|---|---|---|---|---|
| Test MSE | $2.96 \times 10^{-4}$ | $1.56 \times 10^{-5}$ | $3.98 \times 10^{-2}$ | $1.47 \times 10^{-2}$ | $7.85 \times 10^{-4}$ | $5.34 \times 10^{-2}$ |

### E.2 Empirical Risk Optimisation in Generalised Linear Models

This set of examples is based on optimising the empirical risk of a generalised linear model (GLM) with link functions, i.e.,

$$\min_{w} \ \frac{1}{m} \sum_{i=1}^{m} \frac{1}{2} \big( \lambda(w^T x_i) - y_i \big)^2,$$

where $m$ is the number of samples. Wang & Wibisono (2023) show that GLMs with activation functions including leaky ReLU, quadratic, logistic, and ReLU satisfy quasar-convexity.

Each data point $x_i$ is sampled from $\mathcal{N}(0, I_n)$ and the label $y_i$ is generated as $y_i = \lambda(w_*^\top x_i)$, where $w_* \sim \mathcal{N}(0, I_n)$ is the true parameter and $\lambda(\cdot)$ is the link function (sigmoid, ReLU, or leaky ReLU with parameter $\alpha = 0.5$). Due to randomness in the scheme and the initial points, each experiment is run 20 times for both RM and GD, and the average is reported. We consider $m = 1000$ samples in dimension $n = 50$. The initial points are sampled from $10^{-2} \mathcal{N}(0, I_n)$. The parameters $L_1$ and $\gamma$ are unknown, so we tune them numerically. The step size is set to $10^{-2}$ for both algorithms. The number of iterations is $2 \times 10^4$ for the sigmoid link and $6 \times 10^3$ for the other two links. We set $\mu = 10^{-4}$ and $t = 1$. As $t = 1$, the number of gradient calls for GD is the same as the number of iterations. The number of function calls for RM is twice the number of iterations. Figures 8, 9, and 10 show the average objective value and its standard deviation over 20 runs as a function of iterations and CPU time for the sigmoid, leaky ReLU, and ReLU link functions.

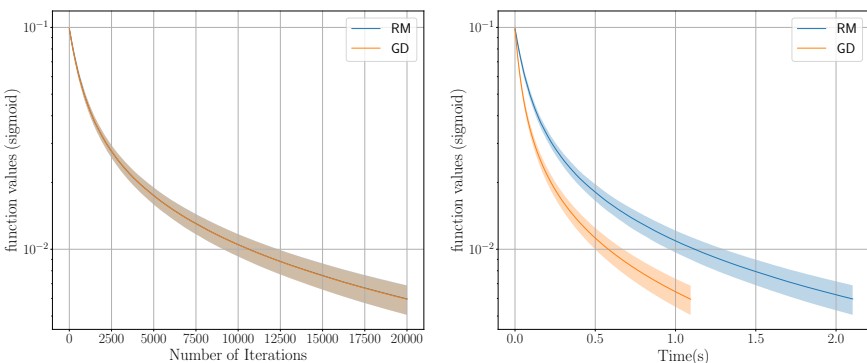

Figure 8: GLM with sigmoid link: average objective value (mean $\pm$ one standard deviation) over 20 runs.

### E.3 Hard Quasar-Convex Function $\bar{f}_{T,\sigma}$

Hinder et al. (2020) introduce a "hard" quasar-convex function

$$\bar{f}_{T,\sigma}(x) = q(x) + \sigma \sum_{i=0}^{T} \Upsilon(x_i),$$

where $\Upsilon(\theta) = 120 \int_1^\theta \frac{t^2(t-1)}{1+t^2} \mathrm{d}t$ and $q(x) = \frac{1}{4}(x_1 - 1)^2 + \frac{1}{4} \sum_{i=1}^{T-1}(x_i - x_{i+1})^2$. They show that $L_1 = 3$ and $\gamma = \frac{1}{100T\sqrt{\sigma}}$ for this function. We choose $T = 20$, $\sigma = 10^{-6}$, $N = 50000$, $h = 10^{-3}$, $\mu = 10^{-4}$, $t = 1$, and sample the initial point from $\mathcal{N}(0, I_T)$. As $t = 1$, the number of gradient calls for GD is the same as the number of iterations. The number of function calls for RM is twice the number of iterations. Figure 11 shows the average objective value and standard deviation over 10 runs versus iterations and CPU time for RM

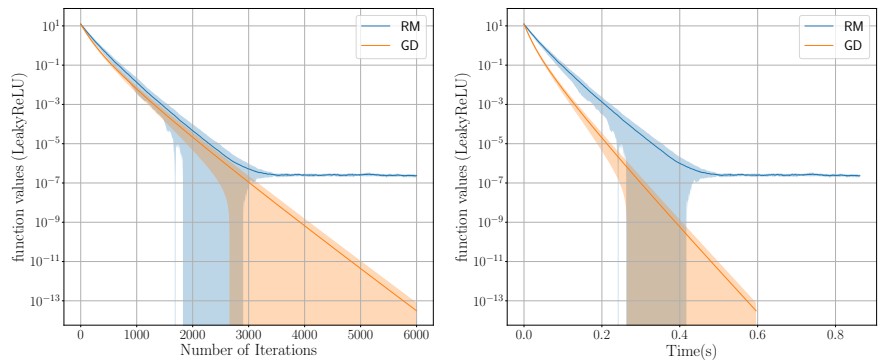

Figure 9: GLM with leaky ReLU link: average objective value over 20 runs.

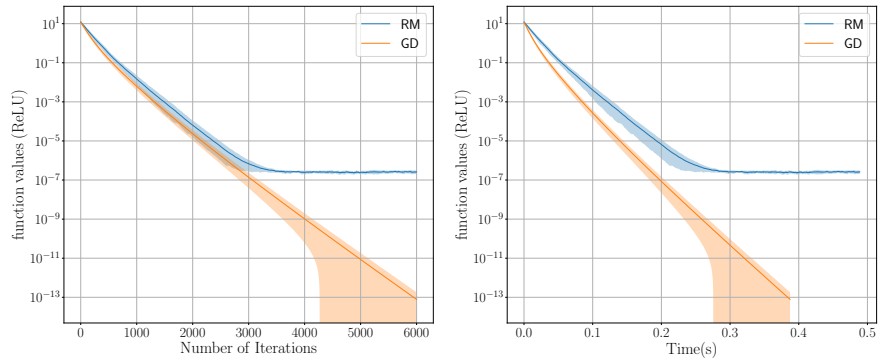

Figure 10: GLM with ReLU link: average objective value over 20 runs.

and GD. We can see that, while having less information, Algorithm 1 performance is similar to GD. In this example, querying the function value is more expensive than calculating the gradient due to the existence of the integral. Thus, each iteration of Algorithm 1 takes more time than each iteration of GD.

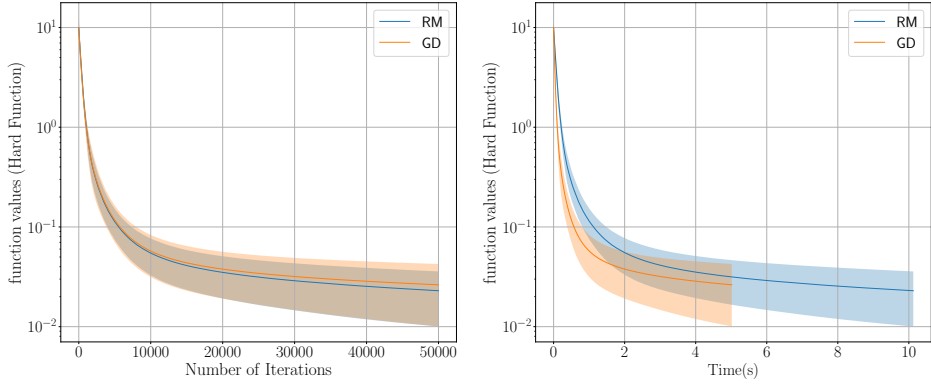

Figure 11: Hard function $\bar{f}_{T,\sigma}$: average objective value (mean $\pm$ one standard deviation) over 10 runs.

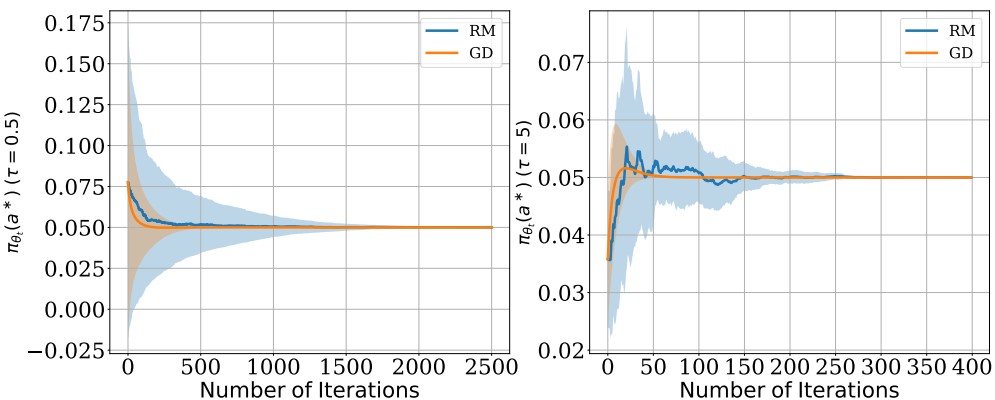

Figure 12: Entropy-regularised bandit: soft sub-optimality error (mean over 20 runs) for two values of $\tau$.

### E.4 Entropy Regularised Policy gradient Reinforcement Learning

We next evaluate Algorithm 1 on an entropy-regularised policy gradient problem. Mei et al. (2020, Lem. 15) shows that the corresponding objective function satisfies the PL inequality and has Lipschitz gradients. Since it has not been shown that the cost function below satisfies quasar-convexity, we apply Algorithm 1 without theoretical convergence and performance guarantees. Following Mei et al. (2020, App. D.2), we consider a single-state bandit with 20 actions. The rewards $r(a) \in [0, 1]$ are chosen randomly, the initial parameter $\theta$ is sampled from a Gaussian distribution, the step size is $h = 0.1$, $\mu = 10^{-5}$, and $t = 1$ (see (Mei et al., 2020, Secs. 2 and 4) for full details). As $t = 1$, the number of gradient calls for GD is the same as the number of iterations and the number of function calls for RM is twice the number of iterations. The loss is

$$\max_{\theta} \ \mathbb{E}_{a \sim \pi_\theta}[r(a) - \tau \log(\pi_\theta(a))]$$

and the convergence metric is the soft sub-optimality error

$$\delta_t = \pi_\tau^{*\top}(r - \tau \log(\pi_\tau^*)) - \pi_{\theta_t}^\top(r - \tau \log(\pi_{\theta_t})).$$

Figure 12 shows the average soft sub-optimality error over 20 runs for $\tau = 0.5$ and $\tau = 5$. We can see that, while having less information, Algorithm 1 performance is similar to GD. Similarly to Cen et al. (2022), increasing $\tau$ leads to smaller final $\delta_t$ and faster convergence.

### E.5 Recurrent Neural Network with Residues

In this example, we evaluate Algorithm 1 on a simple recurrent neural network (RNN) with one hidden layer. The formulation is closely related to the LDSI experiment but now it includes a nonlinear activation function. We consider observations $\{u_i, y_i\}_{i=1}^T \subset \mathbb{R} \times \mathbb{R}$, corresponding to a non-linear model

$$x_{i+1} = \Gamma(Ax_i + Bu_i), \qquad \hat{y}_i = Cx_i + Du_i,$$

where $\Gamma(\cdot)$ is the activation function, $x_i \in \mathbb{R}^n$ is the hidden state, and $A \in \mathbb{R}^{n \times n}$, $B \in \mathbb{R}^{n \times 1}$, $C \in \mathbb{R}^{1 \times n}$, $D \in \mathbb{R}$ are again unknown weights. We minimise the loss

$$\frac{1}{T} \sum_{i=1}^T (y_i - \hat{y}_i)^2,$$

we set $n = 20$ (number of cells in the hidden layer), $T = 100$ (sequence length), and generate 500 sequences and targets sampled from a zero-mean unit-variance normal distribution. The sigmoid function is used as activation. We adopt a full-batch approach and minimise

$$\frac{1}{|\mathcal{B}|} \sum_{(u,y) \in \mathcal{B}} \left( \frac{1}{T - T_1} \sum_{i=T_1}^T (y_i - \hat{y}_i)^2 \right),$$

where $\mathcal{B}$ is a batch of 500 sequences and $T_1 = T/10$. The initial weights are sampled from a zero-mean unit-variance normal distribution. We tune the remaining parameters numerically and set $N = 1000$, $h = 10^{-4}$, $\mu = 10^{-4}$, and $t = 1$. The number of gradient calls for GD is the same as the number of iterations, and the number of function calls for RM is twice the number of iterations. While this network is not proven to satisfy quasar-convexity globally, Hardt & Ma (2017) shows that linear residual networks satisfy PL-type conditions under certain assumptions. In light of Lemma 3 and the LDSI structure, we include this example as an exploratory case. Figure 13 illustrates the network structure, and Figure 14 reports the average objective value and standard deviation over 3 runs for RM and GD as functions of iterations and CPU time. In this example, similar to the results in Section 4.1, under same step-size selection, despite of having less information, RM performs better than GD. A possible explanation for this phenomenon is given in Appendix F.

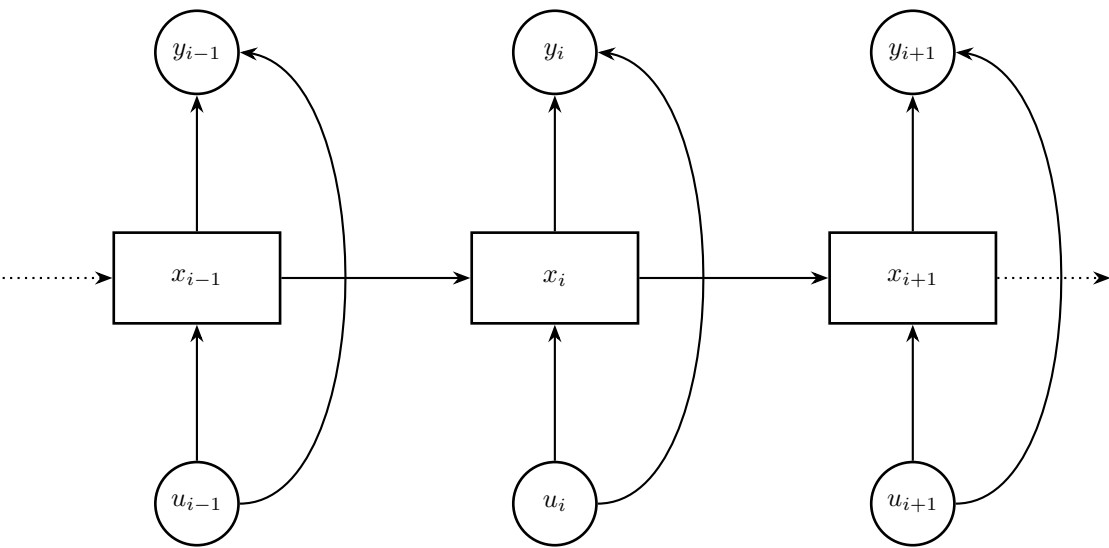

Figure 13: Structure of the trained recurrent network.

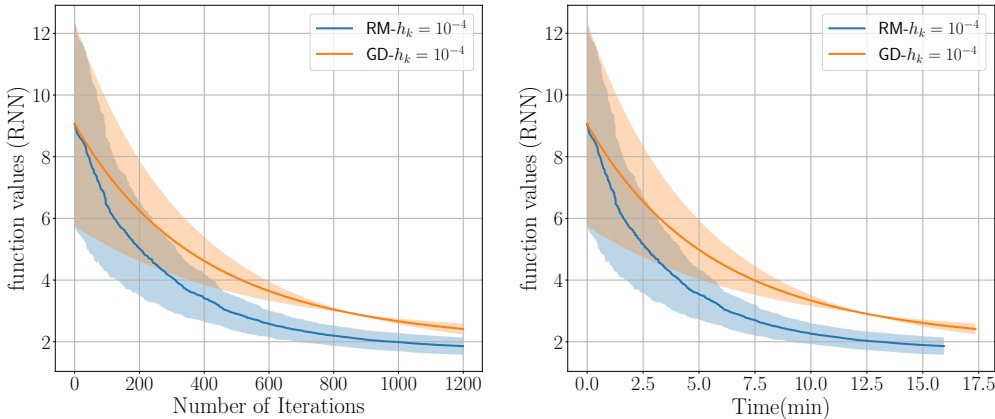

Figure 14: RNN experiment: average objective value (mean $\pm$ one standard deviation) over 3 runs for RM and GD.

### E.6 Constrained Strongly Quasar-Convex Cost Function

In this section, we report experiments on the synthetic strongly quasar-convex test function introduced in Hermant et al. (2024). The objective is of the form

$$h(x) = f(\|x\|_2) \, g\left(\frac{x}{\|x\|_2}\right), \qquad x \in \mathbb{R}^d,$$

with

$$f(t) = t^2, \qquad g(x_1, \ldots, x_d) = \sum_{i=1}^{d} \left(a_i \sin(b_i x_i)^2 + 1\right),$$

and where the coefficients $\{a_i\}_{i=1}^{d}$ and $\{b_i\}_{i=1}^{d}$ are drawn independently according to

$$a_i \sim \mathrm{Unif}[0, 1], \qquad b_i \sim \mathrm{Unif}[-2.5, 2.5].$$

Following Hermant et al. (2024), we fix $d = 100$ in all experiments. This construction yields a 2-strongly-1-quasar-convex function, while exhibiting substantial nonconvexity through the oscillatory $\sin(\cdot)$ terms. The radial factor $f(\|x\|_2) = \|x\|_2^2$ enforces a quadratic growth away from the minimiser, whereas the angular factor $g(x/\|x\|_2)$ introduces many regions of negative curvature along the sphere.

**Constrained variant.** To study projected algorithms, we additionally consider the constrained problem

$$\min_{x \in \mathbb{R}^d} h(x) \quad \text{subject to} \quad \|x\|_1 \leq \rho,$$

for a prescribed radius $\rho > 0$. The feasible set is the $\ell_1$-ball. This setting preserves the nonconvex angular structure of $h$ while imposing a simple convex constraint, and allows for a comparison between projected gradient descent and Algorithm 1. We set $\rho = 20$, $N = 10000$ and $h = 10^{-3}$ for both algorithms and report the results for the average of 10 runs. We test 3 scenarios where i) $t = 1$ (number of sampled directions) and $\mu = 10^{-5}$, ii) $t = 1$ and $\mu = 10^{-10}$, and iii) $t = 10$ and $\mu = 10^{-10}$, without any change in projected GD. The results are shown in Figure 15. It can be seen that the convergence behaviour of Algorithm 1 is consistent with Theorem 4. Decreasing $\mu$ leads to convergence to a point closer to the optimal point and increasing $t_k$ leads to a trajectory closer to GD as the variance of the random oracle is lower. By tuning the parameters $t_k$ and $\mu$, the convergence is more accurate and closer to projected GD.

### E.7 Black-Box Nonlinear System Identification: Details

We provide full details of the black-box nonlinear system identification experiment described in Section 4.2.

The underlying system is a chain of 5 coupled mass-spring-damper oscillators (Figure 7), identical in structure to the system described in Appendix E.1, but augmented with two physically motivated nonlinearities:

- **Actuator saturation.** The input force applied to each mass is restricted to the interval $[-u_{\max}, u_{\max}]$, modelling the finite capacity of a physical actuator, known as *actuator saturation*. This is modelled using the *saturation* function. Specifically,

$$\phi(u) = \mathrm{sat}(u, u_{\max}) = \begin{cases} u_{\max} & u < u_{\max} \\ u & u \in [-u_{\max}, u_{\max}] \\ -u_{\max} & u < -u_{\max} \end{cases}, \tag{39}$$

  where $u_{\max} = 2$. This operation is non-differentiable at $u = \pm u_{\max}$.

- **Coulomb friction.** Each mass experiences a velocity-dependent friction force that opposes its motion with constant magnitude:

$$\psi(v) = -\nu_c \, \mathrm{sign}(v), \quad \nu_c = 0.2. \tag{40}$$

  This operation is discontinuous at $v = 0$.

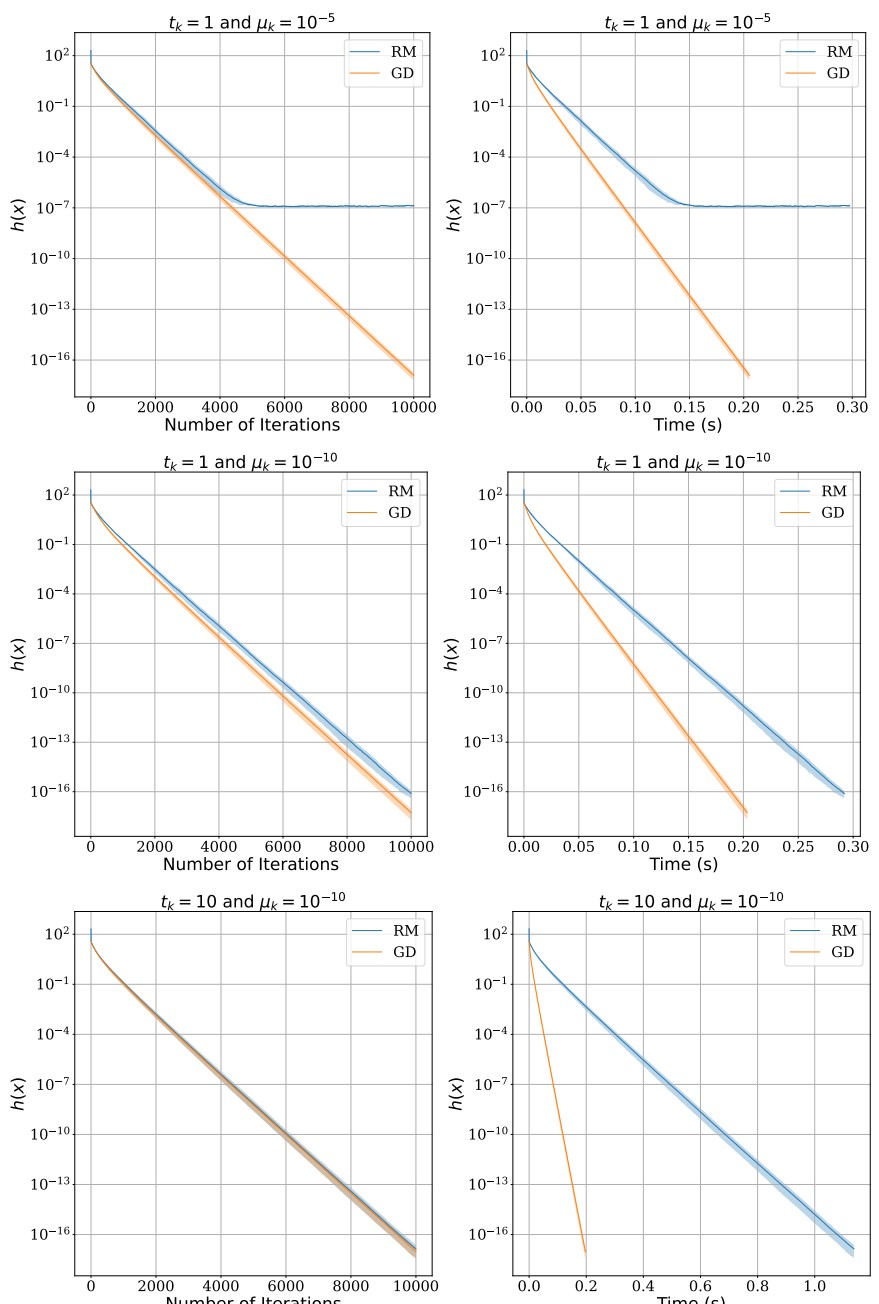

Figure 15: Constrained strong quasar-convex function experiment: average cost function value over 10 runs

The discrete-time dynamics of the true system are given by

$$x_{t+1} = Ax_t + B\,\phi(u_t) + T_s\,\Psi(x_t), \qquad y_t = Cx_t + D\phi(u_t) + \xi_t, \tag{41}$$

where $\Psi(x_t) \in \mathbb{R}^d$ collects the Coulomb friction terms applied to each velocity component, and $\xi_t \sim 10^{-3}\mathcal{N}(0,1)$ is observation noise. The matrices $(A, B, C, D)$ are constructed from the coupled mass-spring-damper chain with parameters identical to those in Appendix E.1. The loss function (17) is evaluated by rolling out the nonlinear dynamics (41) with candidate parameters $(\hat{A}, \hat{C}, \hat{D})$ and comparing the predicted outputs against recorded training data. Since the forward pass involves the unknown non-linear operations,

the gradient of the loss with respect to the parameters cannot be computed. This makes gradient-based optimisation inapplicable and motivates the use of zeroth-order methods.

We generate $N = 200$ input sequences of length $T = 300$, with inputs drawn from $\mathcal{N}(0, 1)$. The output sequences are recorded from the nonlinear simulator (41) with the true parameters $(A, B, C, D)$. The first $T_1 = T/4$ time steps are discarded as a burn-in period. The initial parameters $(\hat{A}_0, \hat{C}_0, \hat{D}_0)$ are obtained by perturbing the true parameters with additive noise of magnitude $\epsilon = 0.1$, with the spectral radius of $\hat{A}_0$ projected to be less than 1. For Algorithm 1, we set $N = 300$, $\mu = 10^{-4}$, $t = 1$, and $h = 10^{-2}$ (fixed step size). For the Armijo line search variant, the maximum step size is $h_0 = 10^{-1}$. For CMA-ES, we set the initial step size $\sigma_0 = 5 \times 10^{-2}$ and use the population size of 18 as suggested by Hansen (2016, eq (48)).

The results in Figure 5 show that Algorithm 1 successfully minimises the black-box objective, converging to a low loss value despite the complete absence of gradient information. The Armijo line search variant achieves faster initial convergence than CMA-ES, while the fixed step-size variant converges steadily.

## F  Possible Explanation for Why Algorithm 1 Outperforms GD in LDSI

While the performance of ZO methods has been generally considered inferior to first-order methods with less information available, our experiments in Section 4.1 and Appendix E.5 have shown that it actually performs *better* than the gradient method when learning a linear dynamical system. In this section, we will examine this phenomenon, which may shed light on future algorithmic development on learning linear dynamical systems and recurrent neural network training.

Let $f \colon \mathbb{R}^n \to \mathbb{R}$ be a twice differentiable function. Recall that each update in Algorithm 1 is given by $x_{k+1} = x_k - h_k g_\mu(x_k)$ for some step size $h_k > 0$ and some gradient approximation $g_\mu(x_k)$ with $\mathbb{E}[g_\mu(x_k)] = \nabla f_\mu(x_k)$. Therefore, given the iterate $x_k$ at the $k$-th iteration, the expected update rule is

$$\mathbb{E}[x_{k+1}|x_k] = x_k - h_k \nabla f(x_k) - h_k(\nabla f_\mu(x_k) - \nabla f(x_k)). \tag{42}$$

A natural question then arises: Where does the vector $-h_k(\nabla f_\mu(x_k) - \nabla f(x_k))$ drift the iterate to and how does smoothing take place in the iterate?

To answer this, let us compute the vector explicitly. Nesterov & Spokoiny (2017) shows that the gradient of the Gaussian smoothed function can be written as

$$\nabla f_\mu(x) = \frac{1}{\kappa} \int_{\mathbb{R}^n} \frac{f(x + \mu u) - f(x)}{\mu} Bu e^{-\frac{1}{2}\|u\|^2} u \, du$$

with $\kappa$ defined in (2). Therefore, using Taylor series, we have

$$
\begin{aligned}
\nabla f_\mu(x) - \nabla f(x) &= \frac{1}{\kappa} \int_{\mathbb{R}^n} \left( \frac{f(x + \mu u) - f(x)}{\mu} - \langle \nabla f(x), u \rangle \right) Bu e^{-\frac{1}{2}\|u\|^2} u \, du \\
&= \frac{1}{\kappa} \int_{\mathbb{R}^n} \left( \frac{f(x) + \mu \nabla f(x)^T u + \frac{\mu^2}{2} u^T \nabla^2 f(\xi(x, u)) u - f(x)}{\mu} - \langle \nabla f(x), u \rangle \right) Bu e^{-\frac{1}{2}\|u\|^2} u \, du \\
&= \frac{\mu}{2\kappa} \int_{\mathbb{R}^n} u^T \nabla^2 f(\xi(x, u)) u Bu e^{-\frac{1}{2}\|u\|^2} u \, du \tag{43}
\end{aligned}
$$

where $\xi(x, u) \in \mathbb{R}^n$ is some vector on the line segment between $x$ and $x + u$; i.e., $\xi(x, u) = tx + (1 - t)(x + u)$ for some $t \in [0, 1]$. Since each $\xi(x, u)$ depends on the direction $u$, it is in general impossible to compute the integral. Having said that, the integral sheds light on the information contained in the vector $\nabla f_\mu(x) - \nabla f(x)$.

For simplicity, let the entries of the random vector in the Gaussian smoothing be independently and identically distributed by assumption, i.e., $B = I_n$. Consider some $u \in \mathbb{R}^n$ and write $\xi = \xi(x, u)$ for simplicity of notation. Suppose that the second derivative is continuous and thus the Hessian $\nabla^2 f(\xi)$ can be diagonalised as $\nabla^2 f(\xi) = \sum_{i=1}^n \lambda_i v_i v_i^T$ for some set of eigenvectors $\{v_i\}_{i=1}^n$ and eigenvalues $\{\lambda_i\}_{i=1}^n$. Moreover, we write the vector $u = \sum_{i=1}^n a_i v_i$ as the linear combination of the basis vectors. Putting aside the probability density, then, each vector $u$ carries a weight of $u^T \nabla^2 f(\xi(x, u)) u = \sum_{i=1}^n \lambda_i a_i^2$. This implies that the integral

$\frac{1}{\kappa}\int_{\mathbb{R}^n} u^T \nabla^2 f(\xi(x,u))uBue^{-\frac{1}{2}\|u\|^2}udu$ takes the average of the landscape of the neighbourhood of $x$, in the sense that more weight is given to the direction whose curvature is large. Contrary to the issues of exploding or vanishing gradients of gradient descent when learning a linear dynamical system, this "stablises" the gradient approximation and therefore reduces the chance of exploding gradients; see Section 4.1 for numerical results. Such inference also applies to other zeroth-order methods, such as uniform smoothing Huang et al. (2022) and coordinate-wise smoothing Chen et al. (2024).

The above development sheds light on the algorithmic design in a recurrent neural network training problem. Having a similar representation as a linear dynamical system identification problem (except for the nonlinear transition of the state), it is known that gradient-based learning algorithms encounter the problem of exploding and vanishing gradients; see, e.g., Bengio et al. (1994); Pascanu et al. (2013); Hardt et al. (2018). As Pascanu et al. (2013) hypothesised that the problem of exploding gradient of recurrent neural network occurs when the curvature along some direction explodes, we see that the expected update of Gaussian smoothing ZO oracle (42) avoids the issue by escaping from such a direction.

