# OpenReview forum: "Minimisation of Quasar-Convex Functions Using Random Zeroth-Order Oracles"
_TMLR — Accepted by TMLR_

### Review · Reviewer_xiPW · 2026-02-19

**Summary Of Contributions:**

This paper studies a random Gaussian smoothing zeroth-order scheme for minimising quasar-convex and strongly quasar-convex functions in
both unconstrained and constrained settings. The paper first considers the unconstrained setting, for which it shows that the random zeroth-order oracle has iteration complexity $O(n\epsilon^{-1})$ for quasar-convex functions and $O(n\log(\epsilon^{-1}))$ for strongly quasar-convex functions. They match the order of the best known iteration complexities for convex and strongly convex functions respectively.  For the constrained setting, the paper introduces introduces proximal quasar-convexity, as an analogue to quasar-convexity in non-smooth optimisation. The paper shows that under proximal quasar-convexity assumption, the projected random zeroth-order oracle achieves $O(n\epsilon^{-1})$ iteration complexity and under strong proximal quasar-convexity, $O(n\log(\epsilon^{-1}))$ iteration complexity. The paper also conducts numerical experiments to show that the random zeroth-order oracle can perform comparably or sometimes even better to the first-order methods.

**Audience:**

Yes

**Audience Explanation:**

I believe so. This is an optimization paper with applications to machine learning problems.

**Claims And Evidence:**

Yes

**Claims Explanation:**

I believe so. In terms of theory, each theoretical result has a rigorous proof, and there is also Appendix C that discusses the other function classes related to the notation of quasar-convexity. The numerical experiments are extensive, which include linear dynamical system, SVM with smoothed hinge loss function, and a few additional examples in the appendix.

**Requested Changes:**

(1) On the page 3, before The Gaussian smoothed version, there should be a dot.

(2) I have some question about $\kappa$ in equation (2). It is defined as $\int_{\mathbb{R}^{n}}e^{-\frac{1}{2}\Vert u\Vert^{2}}du$, which is independent of $B$, which is puzzling where $B$ comes from (unless you replace $-\frac{1}{2}\Vert u\Vert^{2}$ by a more general quadratic form that depends on $B$.

(3) In the paragraph after Remark 2, the stepsize is $h_{k}$ which might depend on $k$. But in your Algorithm 1, it seems you work with a constant stepsize when you update $x_{k+1}$.

(4) I thought the purpose of using zeroth-order oracle is to avoid using the gradient of the objective $f$. However, in your Assumption 1, you need to assume that $f$ is continuously differentiable, even though it seems your Algorithm 1 does not require the differentiability of $f$. Is the assumption of differentiable essential to the proof techniques or is it essential to the design of the algorithm and implementation in practice? If you can relax this differentiability assumption, it would certainly make the paper stronger. If it is unavoidable, it might be worth citing some literature that uses this assumption for zeroth-order oracle methods.

(5) In the references, some names need to be capitalized. For example, in Larry Armijo, lipschitz should be Lipschitz, Pacific Journal of mathematics should be Pacific Journal of Mathematics.

---

> ### Author Response · Authors · 2026-03-16
> **Addressing the issues raised by the reviewer**
>
> Thank you for the precise summary and the valuable comments and suggestions. We have submitted a revised manuscript after addressing all the issues raised by the reviewers. The parts of the revised manuscript that underwent significant changes as compared to our original submission are highlighted in blue. We address your comments below.
>
> **Requested Changes:**
> 1. Thank you for pointing this out. This has been fixed in the revised manuscript.
> 2. You are absolutely right. While we assume the matrix $B$ to be the identity matrix, the norms in (2) should be written as weighted norms as it comes before the assumption. We have clarified this and improved the presentation in the revised manuscript.
> 3. Thank you for pointing this out. Algorithm 1 has been corrected.
> 4. Indeed, the differentiability of the cost function is not required for implementing Algorithm 1. We assume the continuous differentiability of the cost function, because (i) it is a standard assumption in the ZO optimisation literature for deriving theoretical convergence guarantees (see, e.g.,~[1--4]); (ii) the definition of (strong) quasar convexity requires differentiability. We have added a clarification in the revised manuscript.
> 5. Thank you for pointing this out. This has been corrected.
>
> --------------------------------------------------------------------------
>
> [1] Nesterov, Y., \& Spokoiny, V. (2017). Random gradient-free minimisation of convex functions. Foundations of Computational Mathematics, 17(2), 527-566.
>
> [2] Huang, F., Gao, S., Pei, J., \& Huang, H. (2022). Accelerated zeroth-order and first-order momentum methods from mini to minimax optimization. Journal of Machine Learning Research, 23(36), 1-70.
>
> [3] Ghadimi, S., \& Lan, G. (2013). Stochastic first-and zeroth-order methods for nonconvex stochastic programming. SIAM journal on optimization, 23(4), 2341-2368.
>
> [4] Wang, Z., Balasubramanian, K., Ma, S. and Razaviyayn, M., 2020. Zeroth-order algorithms for nonconvex minimax problems with improved complexities. arXiv preprint arXiv:2001.07819.

---

### Review · Reviewer_ebJ7 · 2026-02-23

**Summary Of Contributions:**

This paper analyzes the convergence rate of a zeroth-order optimization algorithm based on Gaussian smoothing for quasar-convex and strongly quasar-convex functions in both unconstrained and constrained settings. The first main result demonstrates that the convergence rate for unconstrained quasar-convex problems is $O(n/\epsilon)$, where $n$ is the number of variables. This matches the established convergence rate for convex objectives. The second result establishes an iteration complexity of $O(n \log(1/\epsilon))$, which is consistent with known results for strongly convex settings. The analysis is further extended to constrained settings, where essentially the same convergence orders are obtained. Numerical experiments were performed on two tasks—linear dynamical system identification and Support Vector Machines (SVM) with smoothed hinge loss—to demonstrate that the proposed zeroth-order approach is competitive with, or even faster than, first-order methods in terms of the number of algorithmic iterations.

**Additional Comments:**

- Please check the beginning of Section 4. Probably, "RM" has been dropped?

- Please check the use of \citet and \citep.

- Equation (24): Better to place \qedhere at the end of the equation.

**Audience:**

Yes

**Audience Explanation:**

Zeroth-order approaches are gaining significant attention within the machine learning community. Furthermore, the numerical examples provided in the paper are highly relevant to ML applications.

**Broader Impact Concerns:**

No concern has been spotted by the reviewer.

**Claims And Evidence:**

Yes

**Claims Explanation:**

As stated in the paper, the authors are the first to establish the convergence rate of a zeroth-order approach for quasar-convex and strongly quasar-convex problems. The rates derived for these function classes match the known results for convex and strongly convex problems. These are mathematically significant results.

**Requested Changes:**

1. While the Big-O notation (e.g., $O(n/\epsilon)$) helps simplify the discussion, it obscures the impact of other critical factors. In particular, the condition number should be explicitly considered throughout the analysis. In Corollary 1, the factor $L_1 / \gamma^2$ appears. The authors should discuss how this compares to standard first-order methods. In Corollaries 2 and 3, a more detailed discussion regarding the parameters $\beta$, $L$, and $\gamma$ is necessary. Again, a formal comparison with first-order methods would help to show the efficiency of the proposed method.

2. The numerical experiments demonstrate an advantage over GD. However, the evaluation is missing comparisons with other widely used zeroth-order algorithms, such as CMA-ES. While it is generally understood that zeroth-order algorithms are slower than gradient-based methods when gradients are available, the fact that the proposed method can outperform GD in certain cases suggests that other derivative-free solvers like CMA-ES might also be highly competitive. Without this comparison, the relative strength of the proposed approach remains unclear.

3. The rationale behind selecting step sizes $h = 10^{-5}$ and $h = 10^{-6}$ for GD is not sufficiently explained. Given that $h = 10^{-5}$ outperformed $h = 10^{-6}$, it is possible that an even larger step size could yield better results. The authors should justify these choices or provide a sensitivity analysis to ensure the GD baseline is properly tuned.

4. The number of samples $t$ used in the RM is not specified. Since RM requires $t$ function calls per iteration whereas GD requires one gradient call, this value is critical for a fair comparison. To make the right-hand panels of the figures meaningful, please report the average and standard deviation of the computational time for both function calls ($f$-calls) and gradient calls ($\nabla f$-calls).

---

> ### Author Response · Authors · 2026-03-16
> **Addressing the issues raised by the reviewer**
>
> Thank you for the precise summary and the valuable comments and suggestions. We have submitted a revised manuscript after addressing all the issues raised by the reviewers. The parts of the revised manuscript that underwent significant changes as compared to our original submission are highlighted in blue. We address your comments below.
>
> **Requested Changes:**
> 1. We have added Remark 3 and Remark 4 following the mentioned corollaries to explicitly compare the dependence of the derived bounds on the problem parameters for Algorithm 1 and standard first-order (FO) methods.
> 2. We have added CMA-ES to the experiment on learning a real dynamical system in Section 4.1 and the experiment on learning the black box nonlinear dynamics in Section 4.2. The parameters of CMA-ES were selected according to [1]. The results show that RM achieves better overall performance compared to the other methods.
> 3. Indeed, alternative step size strategies may improve the performance of the algorithm. To strengthen our experiments, we employed Armijo line search in Section 4.1 and Figure 2, which adaptively selects the largest step size that guarantees a sufficient decrease in the loss function at the next iterate. The maximum allowable step size in the Armijo line search was set to $10^\{-1\}$. We have revised the manuscript and added further explanation.
> 4. Thank you for pointing this out. We have ensured that the number of sampled directions $t$ is specified in all experiments. For RM, the number of function evaluations is twice the number of iterations, whereas for GD, the number of gradient evaluations equals the number of iterations. In addition, we report the CPU (wall-clock) time for all experiments, as an indicative measure under identical computational conditions, to provide a fair comparison that reflects the overall computational cost of the algorithms. We have revised the draft accordingly.
>
> **Additional comments:** Thank you for pointing them out. This has been corrected.
>
> --------------------------------------------------------------------------
>
> [1] Hansen, N. (2016). The CMA evolution strategy: A tutorial. arXiv preprint arXiv:1604.00772.

---

### Review · Reviewer_ARgh · 2026-03-10

**Summary Of Contributions:**

The paper studies gaussian smoothing 0th order optimization for minimizing (strongly) quasar convex functions in unconstrained+constrained settings. It derives upper bounds matching with the corresponding bounds for (strongly) convex functions.

**Audience:**

Yes

**Audience Explanation:**

The paper extends gaussian smoothing for a larger class of functions than previously known.

**Claims And Evidence:**

Yes

**Claims Explanation:**

The paper seems correct with minor clarifications needed.

**Requested Changes:**

There are some clarifications needed for the paper to be technically sound.

 -- def 2: prox quasar functions are definted using parameter $a$, saying $\exists a$ for the definition to hold. In Algorithm 1 and proofs a is set as the step size. Should the definition be amended ? Should it be $\forall a$ which seems restrictive, or should we assume the definition to hold for many $a$ and that the $h$ being used in the algorithm and proofs are a subset of those $a$ ? May be some practical examples of such functions should help along with corresponding $a$ values? the same issue will arise for the strongly quasar version too.

-- i would suggest toning down empirical claims from the proof-of-concepts experiments. also, empirical results could be made stronger by showing how chosen \mu, t  and step-size impact the outcome
minor :

- some citep vs citet issues at several places e.g. Hinder et al on pg 2 and 5
- Algo 1: number of samples are t+1 indexed from 0 to t while the average is calculated over only t samples.
- explicitly write in theorem 4 that it is for \beta-strongly quasar cvx functions

---

> ### Author Response · Authors · 2026-03-16
> **Addressing the issues raised by the reviewer**
>
> Thank you for the precise summary and the valuable comments and suggestions. We have submitted a revised manuscript after addressing all the issues raised by the reviewers. The parts of the revised manuscript that underwent significant changes as compared to our original submission are highlighted in blue. We address your comments below.
>
> **Requested Changes:**
> 1. We have modified Definition 2 and Theorems 3 and 4 statements accordingly. Two examples of functions that satisfy proximal quasar convexity, together with their corresponding parameters, are now provided after Definition 2.
> 2. We have revised the empirical claims of the manuscript accordingly. To strengthen our results, we have added experiments using
>    - different values of $\mu$ and $t$ in Appendix E.6, and
>    - different step-size strategies (both fixed and adaptive) in Section 4.
> 3. Minor issues: Thank you for pointing them out. All have been corrected.

---

### Review · Reviewer_wRbZ · 2026-03-10

**Summary Of Contributions:**

Consider the problem of zeroth-order quasar-convex optimization. This paper proposes solving the problem using projected gradient descent with the “random gradient-free oracle” studied by Nesterov and Spokoiny (2017) and analyzes the iteration complexity for four setups:
1. the objective function is quasar-convex and smooth, and there are no constraints;
2. the objective function is strongly quasar-convex and smooth, and there are no constraints;
3. the objective function is quasar-convex and smooth, the random gradient-free oracle satisfies a variance bound, and there is a nonempty convex constraint set;
4. the objective function is strongly quasar-convex and smooth, the random gradient-free oracle satisfies a variance bound, and there is a nonempty convex constraint set.

**Additional Comments:**

The proof strategies look standard. It would be helpful if the authors could clarify what the primary challenges are.

**Audience:**

Yes

**Audience Explanation:**

The introduction section motivates both quasar-convexity and zeroth-order optimization well.

However, the paper motivates them separately. It would be helpful if the authors could provide some machine learning applications of zeroth-order quasar-convex optimization.

**Broader Impact Concerns:**

This is a paper on optimization theory, and I believe there are no such concerns.

**Claims And Evidence:**

No

**Claims Explanation:**

I have checked the proofs for the first setup and skimmed the remaining proofs. I did not notice any mistakes.

However, the numerical experiments seem inadequate. First, the necessity of solving the two problems—learning dynamical systems and support vector machines—using zeroth-order optimization is unclear. Second, while the paper also studies the strongly quasar-convex case and the constrained optimization case, the two numerical experiments correspond only to the unconstrained quasar-convex optimization case.

**Requested Changes:**

1. Please provide numerical experiments for problems where it is necessary to use zeroth-order optimization.
2. Please provide numerical experiments for the strongly quasar-convex and constrained optimization cases.
3. Please supplement the introduction with motivation for studying zeroth-order quasar-convex optimization problems, in addition to motivating quasar-convexity and zeroth-order optimization separately.
4. If the authors wish to motivate zeroth-order quasar-convex optimization using such arguments, they should explain why this is possible and ensure that the empirical comparisons are fair.

> Although ZO methods are often regarded as inferior to first-order methods in terms of sample efficiency, our numerical results show that the random ZO oracle can perform comparably to, and in some regimes better than, gradient descent when learning a linear dynamical
system, and behaves competitively on other quasar-convex ML tasks such as generalised linear models and support vector machines.

5. In the introduction, the term "random ZO oracle" is used as if it were an algorithm, although it is merely an oracle model. Please correct the terminology.
6. There are some typos. Please correct them. Below are some examples.
    - In the proof of Lemma 1, "the second inequality" and "the third inequality" should be replaced with "the second line" and "the third line," respectively.
    - The first sentence in Remark 2: as ~~it~~ is
    - The period is missing in the second sentence of the first paragraph of Section 2.
    - The period is missing in the last sentence of the paragraph following Definition 2.

---

> ### Author Response · Authors · 2026-03-16
> **Addressing the issues raised by the reviewer**
>
> Thank you for the precise summary and the valuable comments and suggestions. We have submitted a revised manuscript after addressing all the issues raised by the reviewers. The parts of the revised manuscript that underwent significant changes as compared to our original submission are highlighted in blue. We address your comments below.
>
> **Supporting evidence for the claims:**  To motivate the use of ZO methods over quasar-convex functions, we have strengthened the numerical examples by
> - adding an example of a physics-based simulator in Section 4.2, which is completely black-box (with further details given in Appendix E.7),
> - comparing our method with another derivative-free solver, CMA-ES, in Sections 4.1 and 4.2, and
> - clarifying the connection of the examples in Appendix~E.5 and E.6  to (strong) quasar-convex functions.
>
> We hope that these changes will demonstrate the significance of our contributions in ML applications. We address the requested changes below.
>
> **Requested Changes:**
> 1. We have added an example of a physics-based simulator in Section 4.2, in which the gradient information is fundamentally not available. This example shows that our method can effectively identify the parameters of a black-box nonlinear system.
> 2. We have modified the example in Appendix E.6 to capture this setting more prominently in the paper.
> 3. We have revised the introduction of the draft accordingly.
> 4. Thank you for raising this point. Examples where the ZO method outperforms GD include the learning of linear dynamical systems in Section 4.1 and the RNN training experiment in Appendix E.5. It is well known that GD can suffer from gradient vanishing in these settings [1,2], while our experiments suggest that Algorithm~1 may be less affected by this issue. A possible explanation is discussed in Appendix F. In addition, by using step size selection strategies such as Armijo line search and ensuring that all experiments are conducted under identical conditions, we aimed to make the empirical comparisons as fair as possible. The manuscript has been revised to clarify these points.
> 5. Thank you for pointing this out. This has been corrected.
> 6. Thank you for pointing them out. They have been fixed.
>
> **Additional Comments:**
> We leverage standard tools commonly used in the optimisation literature to establish the main results of the paper. One of the primary challenges was to bridge the assumption of (strong) quasar-convexity of the cost function $f$ with the properties of the random oracle $g_\mu$ and the Gaussian-smoothed function $f_\mu$. In particular, the algorithm relies on $g_\mu$ instead of $\nabla f$, while satisfying $E[g_\mu] = \nabla f_\mu$.
>
> Another key challenge was extending the notion of quasar-convexity to a proximal version in order to address constrained minimisation problems. To the best of our knowledge, such an extension has not been previously studied, even for first-order (FO) methods. Based on this extension, we establish the convergence of the sequence generated by Algorithm~1 in the constrained setting.
>
> ---------------------------------------------------------------------------------------
>
> [1] Pascanu, R., Mikolov, T. and Bengio, Y., 2013, May. On the difficulty of training recurrent neural networks. In International conference on machine learning (pp. 1310-1318). Pmlr.
>
> [2] Hardt, M., Ma, T. and Recht, B., 2018. Gradient descent learns linear dynamical systems. Journal of Machine Learning Research, 19(29), pp.1-44.

---

> > ### Comment · Reviewer_wRbZ · 2026-03-16
> >
> > Dear Authors,
> >
> > Thank you for your response. Did you verify the quasar-convexity assumption for the numerical experiment in Section 4.2? I am not sure whether I overlooked that verification.
> >
> > I personally do not consider the proximal version of quasar convexity to be a breakthrough. Such an extension is natural to experts. This does not affect my evaluation, though, according to the TMLR reviewer guidelines. However, it would be helpful to future readers if you could clarify whether there is anything particularly deep in this proximal extension.
> >
> > Best,
> > Reviewer wRbZ

---

> > > ### Author Response · Authors · 2026-03-23
> > > **Addressing the issues raised by the reviewer**
> > >
> > > Thank you for your comment. Since the example in Section 4.2 is a black-box setting that may involve unknown nonlinearity and non-differentiability, an analytical verification of quasar-convexity is not feasible. Nevertheless, we verified the local quasar-convexity of the loss function numerically by sampling 20,000 data points and checking the corresponding inequality, where the directional derivatives were computed numerically. It was observed that $99.96\\%$ of the tested data sequences satisfy the quasar-convexity inequality. For the sake of brevity, we did not include these numerical verification details in the revised manuscript; however, they can be added to the final version if needed.
> > > Moreover, the example in Section 4.1 can also be interpreted as a black-box physics-based simulator, although in this case the underlying process is known to satisfy quasar-convexity. In addition, examples with analytical verification of quasar-convexity are provided in Sections 4.1, 4.3, and Appendices E.1, E.2, E.3, and E.6.
> > >
> > > We also agree that the use of proximal operators is a natural choice when extending the framework to constrained optimisation problems. The constrained setting presented in the paper is included primarily for completeness, as it represents only one component of the overall contributions. Also, the definition of proximal quasar-convexity can be looked at as an extension of quasar-convexity for the non-differentiable cost functions, where the cost function can be decomposed to differentiable and non-differentiable parts, e.g., having $l_1$ regularisation. The theorems in the manuscript only cover the case where the non-differentiable part is an indicator function of a compact convex set (constrained minimisation), but similarly, they can be extended to cover more general cases.

---

### Decision · Action_Editor_xM9M · 2026-05-23

**Recommendation:** Accept as is

**Audience:**

Yes

**Audience Explanation:**

This was a unanimous "Yes" across all four reviewers.

**Claims And Evidence:**

Yes

**Claims Explanation:**

Three out of four reviewers agreed that the theoretical claims are well-supported and rigorously proven. While one of the reviewes initially selected "No" due to a lack of empirical evidence, they acknowledged that the proofs were correct. Furthermore, the authors strengthened the empirical evidence during the rebuttal by adding a black-box physics simulator.